# Applying 3D correlative structured illumination microscopy and X-ray tomography to characterise herpes simplex virus-1 morphogenesis

Kamal L Nahas[1,2], Viv Connor[1], Kaveesha J Wijesinghe[1†], Henry G Barrow[1], Ian M Dobbie[3,4], Maria Harkiolaki[2‡], Stephen C Graham[1]*, Colin M Crump[1]*

[1]Department of Pathology, University of Cambridge, Cambridge, United Kingdom; [2]Beamline B24, Diamond Light Source, Harwell Science and Innovation Campus, Didcot, United Kingdom; [3]Micron Advanced Bioimaging Unit, Department of Biochemistry, University of Oxford, Oxford, United Kingdom; [4]Integrated Imaging Center, Department of Biology, Johns Hopkins University, Baltimore, United States

*For correspondence:
scg34@cam.ac.uk (SCG);
cmc56@cam.ac.uk (CMC)

Present address: †Department of Chemistry, Faculty of Science, University of Colombo, Colombo, Sri Lanka; ‡Department of Chemistry, University of Warwick, Coventry, United Kingdom

Competing interest: The authors declare that no competing interests exist.

## eLife Assessment

This **landmark** manuscript comprehensively examines the roles of nine structural proteins in herpes simplex virus 1 (HSV-1) assembly and nuclear egress. By integrating cryo-light microscopy and soft X-ray tomography, the study presents an innovative approach to investigating viral assembly within cells. The research is thoroughly executed, yielding **exceptional** data that explain previously unknown functions expected to bear widespread influence. This work is of broad interest to virologists, cellular biologists, and structural biologists, offering a robust, contextually rich methodology for studying large protein complex assembly within the cellular environment, serving as an excellent starting point for high-resolution techniques.

**Abstract** Numerous viral genes are involved in the assembly of herpes simplex virus-1 (HSV-1), but their relative importance and function remain poorly characterised. Transmission electron microscopy has been used to study viral protein function in cells infected with HSV-1 mutants; however, these studies were usually conducted without correlative light microscopy to identify specific viral components. In this study, fluorescent capsid (eYFP-VP26) and envelope (gM-mCherry) proteins were imaged by structured illumination microscopy under cryogenic conditions (cryoSIM) and cellular ultrastructure was captured from the same infected cells using cryo-soft-X-ray tomography (cryoSXT). Nine fluorescent HSV-1 mutants, each lacking a different viral protein, were compared to assess the importance of viral proteins in different stages of HSV-1 morphogenesis. The relative importance of five viral proteins to nuclear egress were ranked (pUL34 >pUL21>VP16>pUL16>pUS3) according to the levels of attenuation observed for each virus. Correlative imaging also revealed the roles of five viral proteins in cytoplasmic envelopment. VP16 was found to be important in capsid delivery to envelopment compartments, while cytoplasmic clusters of virus particles plus features of stalled envelopment not previously described were observed in the absence of pUL11, pUL51, gK, and gE. Finally, this 3D imaging approach was used to capture different assembly stages during cytoplasmic envelopment and to determine that envelopment occurs by particle budding rather than wrapping. The findings demonstrate that tomographic 3D correlative imaging is an emerging technology that sheds new light on viral protein functions and virion morphogenesis.

## Introduction

Herpes simplex virus 1 (HSV-1) is a large, enveloped DNA virus from the family *Orthoherpesviridae* that infects mucosal epithelial cells in oral or genital regions and subsequently establishes life-long latent infections in trigeminal and sacral ganglia, respectively (*Nicoll et al., 2012*; *Gatherer et al., 2021*). HSV-1 is composed of three layers: a proteinaceous capsid that contains the DNA genome, an amorphous layer of at least 23 different proteins known as the tegument, and a lipid bilayer envelope studded with viral glycoproteins (*Hernández Durán et al., 2019*; *Owen et al., 2015*; *Ahmad and Wilson, 2020*; *Mettenleiter, 2002*; *Johnson and Baines, 2011*; *Crump, 2018*). Virion assembly involves a complex multi-step pathway in the cell, beginning with the expression of immediate-early viral genes (*Triezenberg et al., 1988*). This is followed by replication of viral DNA, assembly of capsids, and packaging of DNA genomes to form nucleocapsids, all within the nucleoplasm (*Brown and Newcomb, 2011*). Given that nucleocapsids are too large to pass through the channels of nuclear pore complexes, they cross the nuclear envelope in a process known as nuclear egress. This begins with the budding of nucleocapsids into the perinuclear space through the inner-nuclear membrane (primary envelopment). Once in the perinuclear space, the nucleocapsids possess a temporary envelope, acquired from the inner-nuclear membrane, which will then fuse with the outer-nuclear membrane to release the capsid into the cytoplasm (*Bigalke and Heldwein, 2015a*; *Bigalke and Heldwein, 2015b*; *Bigalke and Heldwein, 2016*; *Zeev-Ben-Mordehai et al., 2015*). Tegument proteins begin being deposited on the nucleocapsid in the nucleus and this process continues in the cytoplasm (*Owen et al., 2015*). Cytoplasmic nucleocapsids undergo (secondary) envelopment at cytoplasmic endomembranes, such as *trans*-Golgi network vesicles and/or recycling endosomes (*Ahmad and Wilson, 2020*; *Hollinshead et al., 2012*), hereafter referred to as cytoplasmic envelopment. This results in the enveloped virion being located in the lumen of a carrier vesicle, which fuses with the plasma membrane to release the virion (*Owen et al., 2015*).

The large DNA genome of HSV-1 encodes at least 84 proteins, many of which have uncharacterised functions in virus assembly and differ in the degree of their importance during morphogenesis (*Ahmad and Wilson, 2020*). HSV-1 mutants lacking expression of specific viral proteins or domains thereof have been used to study the role of viral genes in assembly using 2D transmission electron microscopy (TEM). Several phenotypes associated with attenuation have been observed with this technique, including varying numbers of capsids between the nucleus and cytoplasm, clusters of perinuclear virus particles, and stalled envelopment of cytoplasmic nucleocapsids (*Baines and Roizman, 1992*; *Fulmer et al., 2007*; *Le Sage et al., 2013*; *Sarfo et al., 2017*; *Gao et al., 2020*; *Mossman et al., 2000*). However, the extensive sample processing associated with TEM (i.e. chemical fixation, dehydration, hardening by resin-embedding or high-pressure freezing, sectioning, and staining) can distort ultrastructure and complicate the interpretation of features that appear to show attenuation in virus assembly (*Gunning and Calomeni, 2000*; *Bråten, 1978*). Thin sectioning also limits our understanding of the 3D nature of phenotypes associated with attenuation in virus assembly (*Schauflinger et al., 2013*). Volumetric ultrastructural imaging of numerous HSV-1 mutants in near-native conditions is needed to draw direct comparisons between viral proteins and to understand the 3D nature of stalled assembly.

In this study, the roles of nine HSV-1 genes in virus assembly were investigated using an emerging 3D correlative imaging strategy consisting of structured illumination microscopy on cryopreserved samples (cryoSIM) and cryo-soft-X-ray tomography (cryoSXT) (*Kounatidis et al., 2020*). HSV-1 assembly intermediates have previously been observed by cryoSXT without performing cryoSIM, which limited characterisation of virus assembly (*Nahas et al., 2022a*). Using this correlative light X-ray tomography (CLXT) approach, we employed recombinant viruses expressing fluorescently labelled viral proteins of the capsid (VP26) and the envelope glycoprotein M (gM) to distinguish between unenveloped and enveloped particles in order to study different stages of virion assembly (*Douglas et al., 2004*; *Albecka et al., 2016*; *Scherer et al., 2021*). Our study provides an extensive and comparative ultrastructural analysis of herpesvirus morphogenesis, revealing new insights into the virion assembly pathway of these structurally complex viruses.

## Results

### Generation of mutants to study HSV-1 assembly

HSV-1 particles contain three layers: capsid, tegument, and envelope. A dual-fluorescent KOS strain HSV-1 mutant containing eYFP-tagged capsids and mCherry-tagged envelopes — hereafter referred to as the dual-fluorescent parental (dfParental) virus — was used to enable the independent identification of capsids and envelope and thus distinguish between unenveloped and enveloped particles. The tagged virus encodes eYFP conjugated to the N-terminus of the small capsid protein VP26 (*Douglas et al., 2004*; *Albecka et al., 2016*; *Scherer et al., 2021*). This tagging theoretically allows identification of all virus particles except capsid-free 'light' particles (*Heilingloh and Krawczyk, 2017*) in the cytoplasm and immature procapsids in the nucleus that have yet to acquire VP26 (*Chi and Wilson, 2000*). The C terminus of the envelope glycoprotein gM was conjugated to mCherry, enabling identification of gM-containing endomembranes plus viral envelopes (*Scherer et al., 2021*; *Figure 1A*). The dfParental virus was used as the template strain to generate nine HSV-1 mutants lacking specific HSV-1 structural proteins. Mutants were generated to study phenotypes associated with an attenuation of virus assembly caused by the loss of specific viral proteins (*Figure 1A*).

Nine mutants containing single-gene 'knockouts' were generated either by introducing stop codons or by sequence deletion (*Figure 1B*). As pUL34, VP16, and gK are all essential for virion assembly, the ΔUL34, ΔVP16, and ΔgK viruses were cultured using complementing cell lines to generate infectious virions that carry the corresponding protein but are unable to synthesise it, such that virus entry and gene expression could occur as normal, but the protein would be absent during virus assembly. Protein levels of the corresponding knockout genes were undetectable in Vero cells infected with each of the HSV-1 mutants by immunoblot (*Figure 1C–D*). No suitable antibody was available to detect gK expression and so the presence of the inserted stop codons was confirmed by PCR amplification of the relevant region of the HSV-1 genome and Sanger sequencing (*Figure 1—figure supplement 1A–B*). Furthermore, immunoblotting for pUL20, the obligate binding partner of gK, revealed substantially reduced pUL20 expression in cells infected with the ΔgK mutant (*Figure 1C*) as previously observed for independent gK deletion mutants (*Foster et al., 2004*). The intensity of the pUS3 bands were reduced for the ΔpUL21 and ΔUL34 mutants (*Figure 1C*). The US3 gene has a propensity to mutate in response to pUL21 deletion (*Benedyk et al., 2021*), but this was not known at the time these virus stocks were generated, and the ΔpUL21 virus was not grown in a complementing cell line.

### Replication kinetics and cell-to-cell spread of HSV-1 mutants

Replication kinetics for all the dual-fluorescent viruses were compared over 24 hr in single-step growth curves (*Figure 2*, *Figure 2—figure supplement 2*). These data were compared with measurements of plaque size 72 hr post-infection (hpi) that served as collective indicators of attenuation in both replication and cell-to-cell spread (*Figure 2B–C*). In line with previous work showing that gE is more important in cell-to-cell spread than in virus assembly (*Johnson and Baines, 2011*; *Polcicova et al., 2005*; *Wisner and Johnson, 2004*; *Farnsworth et al., 2003*), defects in replication of the ΔgE virus were not detected in the replication curves (*Figure 2A*), but the plaque sizes of this mutant were significantly reduced compared with dfParental (*Figure 2B–C*). gE interacts with a complex of the tegument proteins pUL11, pUL16, and pUL21, providing one route to link the tegument and envelope layers during virus assembly (*Han et al., 2012*; *Klupp et al., 2005*; *Meckes et al., 2010*; *Yeh et al., 2011*; *Benedyk et al., 2022*). Replication of the ΔpUL11, ΔpUL16, and ΔpUL21 mutants was generally reduced with respect to dfParental HSV-1 (10–100-fold reduction), suggesting these proteins have more important roles than gE during virion assembly and could perform these roles as a complex (*Figure 2A*). ΔpUL11 and ΔgE plaques were significantly smaller than the ΔpUL16 and ΔpUL21 plaques, suggesting the interactions formed by pUL11 and gE are more important for cell-to-cell spread (*Figure 2B–C*; *Farnsworth et al., 2007*). However, viruses lacking functional pUL21 are known to form extremely small plaques and compensatory mutations in the US3 gene can arise when pUL21 is inactivated (*Benedyk et al., 2021*). The sequence of US3 in the ΔpUL21 virus was, therefore, analyzed by PCR and Sanger sequencing, revealing that 84.3% of ΔpUL21 genomes encoded an amino acid substitution (C456T) in the US3 gene (*Figure 2—figure supplement 2A–C*). Residues with similar physicochemical properties (e.g. serine) are present at this position in US3 from other alphaherpesviruses, suggesting that this substitution is unlikely to severely alter US3 activity (*Figure 2—figure supplement 2D–E*). Although the residue is surface exposed (*Figure 2—figure supplement 2F*), it



**Figure 1.** Characterisation of HSV-1 mutants. (**A**) Schematic of HSV-1 assembly with the proposed roles of viral proteins at different stages. Question marks denote that the role of the corresponding protein at that stage of HSV-1 assembly remains uncertain. Mutants with stop codons are named *Δ*+protein name (e.g. ΔpUL11), whereas mutants with sequence deletions are named *Δ*+ gene name (e.g. ΔUL34). Unique long (UL) and unique short (US) names are used except for proteins more commonly known by another name (i.e. ΔVP16, ΔgK, and ΔgE). (**B**) Schematic of recombinant viruses generated. Deletions (black bars) or stop codons (red arrows) were introduced into genes of interest (cyan) to prevent protein expression. Numbering refers to the amino acid residues of the corresponding protein. Flanking genes (blue and green) and pseudogene (mutated US9; grey) ***Negatsch et al., 2011*** in HSV-1 strain KOS are indicated. (**C–D**) Absence of protein expression was confirmed by immunoblotting infected Vero cell lysates with VP5 and GAPDH as viral and cellular loading controls, respectively. Due to the unavailability of an antibody that recognises gK, immunoblotting of pUL20 was used as an indicator of loss of gK expression since stable expression of gK and pUL20 relies on the presence of each other (***Foster et al., 2003***).

The online version of this article includes the following source data and figure supplement(s) for figure 1:

**Source data 1.** Original files of the full raw uncropped, unedited Western blots shown in ***Figure 1C and D***.

*Figure 1 continued on next page*

*Figure 1 continued*

**Source data 2.** Uncropped blots with the relevant bands clearly labelled relating to Western blots shown in *Figure 1C and D*.

**Figure supplement 1.** Sequence alignment of UL53 gene in ΔgK and dfParental HSV-1.

is not located near the active site, suggesting it is unlikely to affect kinase activity (*Figure 2—figure supplement 2G*). Nonetheless, potential alterations to pUS3 activity in the ΔpUL21 mutant should be taken into account when the results for this mutant are interpreted.

After capsid assembly in the nucleus, capsids migrate into the cytoplasm by budding into the perinuclear space through the inner-nuclear membrane, forming a temporarily enveloped particle (*Pražák et al., 2024*). The envelope of the perinuclear particles fuses with the outer nuclear membrane to release the capsid into the cytoplasm. pUL34 is required for budding into the perinuclear space, and pUS3 is thought to regulate this process via phosphorylation of the pUL31/pUL34 nuclear egress complex (NEC) (*Draganova et al., 2021*; *Muradov et al., 2021*; *Thorsen et al., 2021*). While replication of the ΔUS3 virus was 10-fold reduced with respect to dfParental at 24 hpi, the plaques formed by the ΔUS3 mutant were closer in size to dfParental plaques than any other mutants at 72 hpi, suggesting that pUS3 had the least important role of all nine mutants for capsid migration into the cytoplasm and subsequent viral spread (*Figure 2A–C*). In contrast, the replication kinetics of the ΔUL34 mutant were approximately $10^5$-fold reduced, and the mutant did not form plaques, which was expected for an essential component of the NEC (*Figure 2A–C*; *Bigalke and Heldwein, 2015a*; *Bigalke and Heldwein, 2015b*).

VP16, pUL51, and gK are highly important for cytoplasmic envelopment of HSV-1 capsids (*Crump, 2018*; *Lau and Crump, 2015*; *Butt et al., 2020*; *Albecka et al., 2017*; *Svobodova et al., 2012*). By 24 hpi, replication of the ΔpUL51 virus was 10–500-fold reduced and plaque sizes were greatly reduced as well (*Figure 2A–C* & *Figure 2—figure supplement 1*). Replication kinetics of the ΔgK virus were $10^3$–$10^5$-fold reduced at 24 hpi, and no plaques were visible at 72 hpi (*Figure 2A–C* & *Figure 2—figure supplement 1*). This suggests gK is important in cell-to-cell spread as well as virion assembly. It is possible that infectious ΔgK virions were produced but unable to egress from the cell. In this scenario, infectious particles could be detected in cell lysates generated for titration of the replication curves despite an inability of ΔgK HSV-1 to form plaques. In contrast to the ΔgK mutant, the ΔVP16 mutant produced detectable plaques by 72 hpi even though infectious virions could not be detected by 24 hpi with the replication curves, suggesting the lack of VP16 could delay to beyond 24 hr the production of detectable titres of infectious virus (*Figure 2A–C*). Alternatively, it is possible that loss of VP16 leads to low levels of cell-cell fusion to form small syncytia, enabling cell-to-cell spread of viral genome and subsequent plaque formation (*Mossman et al., 2000*).

## Correlative light X-ray tomography of HSV-1 assembly

After generating and characterising the mutants, defects in virus assembly were explored with high-resolution imaging techniques under cryogenic conditions. Cryogenic imaging using soft-X-ray tomography captures the ultrastructure of the cell in 3D and has the advantage of allowing the study of samples in a near-native state without the need for chemical fixation (*Nahas et al., 2022a*; *Chen et al., 2022*). U2OS osteosarcoma cells were used in this study because they have been used for ultrastructural HSV-1 research and have been demonstrated to be durable under X-ray exposure (*Figure 3A*; *Nahas et al., 2022a*; *Simpson-Holley et al., 2005*; *Deng et al., 2014*). U2OS cells were infected at MOI = 2 with dfParental or mutants for 15.5 hr and were stained with MitoTracker Deep Red (Thermo Fisher Scientific) for 30 min to label the mitochondria (*Okolo et al., 2021*). Gold fiducials were added to the surface of the cells to facilitate alignment of tomographic projections (*Okolo et al., 2021*). Samples were cryopreserved by plunge cryocooling in liquid ethane at 16 hpi. In a synchronously infected population, infected cells will progress through stages of virus assembly at different rates (*Nahas et al., 2022a*; *Scherer et al., 2021*; *Drayman et al., 2019*). Cells were cryopreserved at a relatively late timepoint (16 hpi) to increase the proportion of cells at late stages of infection. Infected cells were first imaged by cryoSIM to capture viral eYFP-VP26 and gM-mCherry fluorescence plus MitoTracker mitochondrial fluorescence (*Figure 3B*; *Kounatidis et al., 2020*; *Vyas et al., 2021b*). These same infected cells were later imaged by cryoSXT. Given that mitochondria produce high contrast in cryoSXT tomograms and are easily distinguishable based on their complex shapes and arrangements (*Nahas et al., 2022a*), the MitoTracker stain in the cryoSIM and the mitochondria



**Figure 2.** Replication kinetics and plaque assays of HSV-1 mutants. (**A**) Single-step replication curves on U2OS cells infected at MOI = 2 with virus. dfParental refers to eYFP-VP26 & gM-mCherry KOS used as a parental strain. U2OS cells were infected at MOI = 2 with virus over a 24 hr period and were treated with citric acid at the 1 hr timepoint to deactivate residual input virus. Titrations were performed on parental or complementing Vero cells. Two technical repeats were measured for each timepoint, and the data are representative of two biological replicates (*Figure 2—figure supplement 1*). Error bars show mean ± range. (**B**) 72 hr plaques were immunostained for gD using an antibody conjugated to horseradish peroxidase and were subsequently stained with DAB. Plaque area (pixels) were measured by applying thresholds to intensity using Fiji and quantifying the number of pixels in each plaque from binary masks (*Rueden et al., 2017*; *Schindelin et al., 2012*). Measurements were taken from a range of plaques for each recombinant virus: dfParental (N=50), ΔUS3 (N=44), ΔpUL21 (N=22), ΔpUL16 (N=36), ΔpUL11 (N=31), ΔgE (N=12), ΔpUL51 (N=50), and ΔVP16 (N=50). Given the skewed distributions, non-parametric Mann-Whitney *U* tests were used to assess the significance of differences. P-value thresholds: <0.05 (*), <0.005 (**), and <0.0005 (***). NS, no significance; NA, not applicable. (**C**) Images of 72 hr plaques from dfParental and mutants.

The online version of this article includes the following figure supplement(s) for figure 2:

*Figure 2 continued on next page*

Figure 2 continued

**Figure supplement 1.** Additional replicate for single-step replication kinetics of mutant and dfParental HSV-1.

**Figure supplement 2.** Sequence alignment of US3 in recombinant and dfParental viruses.

in the tomograms were used as landmarks to guide 3D image alignment for correlation (**Kounatidis et al., 2020**). CLXT was used to identify virus particles at various stages of assembly. The nucleus is the site of capsid assembly and nuclear capsids have previously been observed in the tomograms as dark puncta (**Nahas et al., 2022a**). These capsids correlated with the fluorescently tagged capsid protein eYFP-VP26 (**Figure 3C**). gM-mCherry$^+$ vesicles and virus particles were detected in the cytoplasm, enabling the study of cytoplasmic envelopment. Finally, eYFP-VP26+ and gM-mCherry+ particles were detected in spaces between cells, allowing study of extracellular virions (**Figure 3C**).

## Nuclear egress attenuation of ΔpUL16, ΔpUL21, ΔUL34, ΔVP16, and ΔUS3 HSV-1

Nuclei of samples infected with dfParental, ΔpUL16, ΔpUL21, ΔUL34, ΔVP16, and ΔUS3 viruses were labelled with Hoechst stain to distinguish between nuclear and cytoplasmic capsids by cryoSIM (**Figure 4A**). The plasma membrane of infected cells was delineated by digitally oversaturating the gM-mCherry fluorescence. eYFP-VP26 signal was manually thresholded to filter out background and include pixels containing individual or clustered puncta that represent capsids. The number of eYFP-VP26$^+$ pixels in maximum Z projections of the cryoSIM data were quantitated using these nuclear and plasma membrane borders. The nuclear:cytoplasmic (N:C) ratio of capsids was used as measures of nuclear egress attenuation, which could manifest from a defect in nuclear egress or a delay in the replication kinetics of mutants (**Figure 4B**). Infected cells were included in the analysis if the nucleus and cell borders could be easily distinguished, if most of the cell was in the field of view, and if most of the gM-mCherry$^+$ areas of the cytoplasm did not overlap with the nucleus. However, partial overlap between gM-mCherry fluorescence and the nucleus was common (e.g. ΔpUL21 and ΔUL34 in **Figure 4C**), which could have produced an overestimated N:C ratio.

These data demonstrated that ΔpUL16, ΔpUL21, ΔUL34, ΔVP16, and ΔUS3 viruses experienced a defect or a delay in nuclear egress when compared with the dfParental virus (**Figure 4B–C**). The ΔgE virus was included in the analysis as a negative control because this virus did not have delayed replication kinetics (**Figure 2A**), and gE is not suspected to be involved in nuclear egress. The ΔgE virus-infected cells were not stained with Hoescht, and instead, the nucleus was identified using images of nuclei collected from tiled X-ray projection images (X-ray mosaics). The N:C ratio of the dfParental (N=14) and ΔgE (N=12) viruses did not differ significantly, suggesting gE does not play a role in nuclear egress.

pUL34 is essential for nuclear egress and served as a positive control for nuclear capsid retention in this study (**Bigalke and Heldwein, 2015a**; **Bigalke and Heldwein, 2015b**). The ΔUL34 virus had the highest N:C ratios (N=10; **Figure 4B–C**), which was commensurate with the high level of attenuation observed for this virus in terms of replication kinetics and its inability to form plaques (**Figure 2A–C**). For the other mutants, significance in difference was assessed with respect to the N:C ratios of both the dfParental and ΔUL34 viruses. Although the N:C ratios of the ΔUS3 virus (N=13) were significantly higher than those of the dfParental virus, they were also significantly lower than those of the ΔUL34 virus. The ΔUS3 virus had the lowest N:C ratios of the mutants other than the ΔgE negative control (**Figure 4B–C**), suggesting ΔUS3 was least attenuated in nuclear egress, which is commensurate with its modest attenuation in the replication curves and plaque size assay (**Figure 2A–C**).

ΔpUL16 (N=13) and ΔpUL21 (N=11) N:C ratios were significantly higher than those of the dfParental virus but were not significantly different from those of the ΔUL34 virus (**Figure 4B–C**), suggesting a greater defect or delay in nuclear egress than observed for ΔUS3 and consistent with known roles for both proteins in nuclear egress of HSV-2 (**Le Sage et al., 2013**; **Gao et al., 2020**; **Gao et al., 2017**).

N:C ratios of ΔVP16 (N=24) were also significantly greater than those of dfParental, suggesting a defect or delay in nuclear egress (**Figure 4B–C**), consistent with TEM studies implicating VP16 in nuclear egress (**Mossman et al., 2000**; **Naldinho-Souto et al., 2006**). CryoSIM data from infected cells were correlated onto X-ray tomograms, revealing an absence of, or reduction in, cytoplasmic capsids for each of the five mutants (**Figure 4D**).

**Figure 3.** Correlative imaging workflow to study dfParental HSV-1 assembly. U2OS cells were infected at MOI = 2 with dfParental HSV-1 for 16 hr. (**A**) A schematic of sample preparation. U2OS cells were seeded on 3 mm TEM grids, infected with the dfParental virus for 15.5 hr, mitochondria were stained with MitoTracker Deep Red (Thermo Fisher Scientific) for 0.5 hr, and 200 nm gold fiducials were overlayed onto the cells immediately before cryopreservation by plunge cryocooling in liquid ethane. (**B**) Cryopreserved samples were imaged first by cryoSIM and subsequently by cryo-soft-X-ray tomography (cryoSXT). The left and right sides of the cryoSIM image show the data at conventional resolution (left) and after it was super-resolved in a cryoSIM reconstruction (right). CryoSIM fluorescence was then correlated onto the CryoSXT datasets by comparing the MitoTracker stain with mitochondria in the tomograms. Scale bars = 10 μm. (**C**) CLXT was used to identify virus assembly intermediates in U2OS cells infected with the dfParental tagged virus. eYFP-VP26$^+$/gM-mCherry$^-$ particles were identified in the nucleus (N). eYFP-VP26$^+$/gM-mCherry$^+$ particles were identified in the cytoplasm (C) where cytoplasmic envelopment occurs. eYFP-VP26$^+$/gM-mCherry$^+$ particles were also identified in spaces between cells. LD = lipid droplets.



**Figure 4.** CryoSIM and correlative light X-ray tomography (CLXT) of nuclear egress attenuation. U2OS cells were infected at MOI = 2 with indicated viruses for 16 hr. Mutant-specific defects in nuclear egress were investigated using maximum Z projections of cryoSIM data. (**A**) eYFP-VP26 fluorescence was captured in punctate form, representing individual virus particles or clusters. Digitally saturated gM-mCherry was used to delineate the plasma membrane (PM) of infected cells and the Hoechst stain was used to delineate the nucleus (N). A binary mask of eYFP-VP26 fluorescence was generated to include capsids or capsid clusters and filter out background or noise. The arrow indicates fluorescent viral proteins in the cytoplasm. Scale bars = 10 μm. (**B**) The number of pixels containing capsids or capsid clusters in the nucleus and cytoplasm was counted using the plasma membrane and nucleus borders. The nuclear:cytoplasmic (N:C) ratio of capsids was lowest for the dfParental-infected cells and the ΔgE-infected cells (negative control) and was significantly higher for the other mutants. Owing to the skewed distributions, the significance of differences was assessed using non-parametric Mann-Whitney *U* tests between dfParental (N=14) (green statistics) or ΔUL34 (N=10) (red statistics) and other viruses, specifically ΔpUL16 (N=13), ΔpUL21 (N=11), ΔUS3 (N=13), ΔVP16 (N=24), and ΔgE (N=12). NS; no significance. P-value thresholds: <0.05 (*), <0.005 (**), and <0.0005 (***). (**C**) Representative viral fluorescence cryoSIM data for the mutants. Note that the ΔUS3 example was reused in *Figure 5C*. Scale bars = 10 μm. (**D**) Correlative cryoSIM and cryo-soft-X-ray tomography (cryoSXT) data for the mutants. Inset images show correlated nuclear capsids at twice the magnification. Scale bars = 1 μm. Nuclear clusters of capsids (known as assemblons *Ward et al., 1996*; *Lee et al., 2006a*) are visible in the ΔUS3 and ΔpUL16 datasets (arrows). Stars indicate gold fiducials. C, cytoplasm.

## Differential capsid clustering and gM-mCherry⁺ endomembrane association of ΔpUL11, ΔpUL51, ΔgE, ΔgK, and ΔVP16 mutants

CryoSIM imaging showed that interspersed eYFP-VP26+ capsids and gM-mCherry+ endomembranes could be observed at juxtanuclear assembly compartments (JACs) in cells infected with dfParental (*Figure 5A*). Varying degrees of attenuation in cytoplasmic envelopment have been previously reported for HSV-1 mutants lacking some of the tegument proteins or glycoproteins investigated in this study (i.e. pUL11 *Baines and Roizman, 1992*; *Fulmer et al., 2007*; *Leege et al., 2009*, VP16 *Mossman et al., 2000*, pUL51 *Butt et al., 2020*; *Albecka et al., 2017*; *Oda et al., 2016*, gK *Lau and Crump, 2015*; *Hutchinson and Johnson, 1995*, and gE *Farnsworth et al., 2003*; *Farnsworth et al., 2007*), and some proteins have been proposed to participate in cytoplasmic envelopment (i.e. pUL16 *Han et al., 2012* and pUL21 *Finnen and Banfield, 2018*; *Shahin et al., 2017*). For cells infected with ΔpUL11, ΔpUL51, ΔgK, or ΔgE, clusters of capsids could be observed at the JACs (*Figure 5C*). Smaller clusters of capsids were observed in the JACs for the ΔpUL11 and ΔgE mutants (*Figure 5C*), whereas more extensive clusters of capsids were observed in the JACs for the ΔpUL51 and ΔgK. Capsid clustering was not observed in cells infected with ΔVP16 virus, and cytoplasmic capsids appeared to associate less with gM-mCherry+ endomembranes compared with capsids of dfParental, suggesting VP16 is important in capsid recruitment to envelopment compartments (*Figure 5B–C*). The spatial distribution of capsids in the cytoplasm of cells infected with ΔpUL16 and ΔpUL21 could not be reliably assessed due to a paucity of cytoplasmic capsids detected, presumably arising from their observed defect/delay in nuclear egress (*Figure 4B–D*).

To quantitate cytoplasmic distribution of virus particles, the intensity of gM-mCherry fluorescence was saturated and used to delineate the approximate borders of JACs (*Figure 5A*, *Figure 5—figure supplement 1A*). dfParental capsids in these assembly compartments associated closely with gM-mCherry⁺ endomembranes. However, cells infected with the ΔVP16 virus contained fewer capsids in the JAC and they were less closely associated with gM-mCherry⁺ endomembranes (*Figure 5A–C* & *Figure 5—figure supplement 1B*). Thresholds on intensity were applied to the eYFP-VP26 and gM-mCherry cryoSIM fluorescence to produce binary masks where noise and background were filtered out. The ratio of eYFP-VP26⁺ pixels to gM-mCherry⁺ pixels in the binary masks was measured for each JAC and were significantly lower for the ΔVP16 virus when compared with dfParental (*Figure 5D*), indicating fewer capsids at the JAC for this virus. This is consistent with the lack of overlap between capsids and gM-mCherry+ endomembranes observed (*Figure 5B*) plus the observed defect/delay in nuclear egress (*Figure 4B–D*). ΔUS3 was included in the analysis as a negative control for a virus with impaired nuclear egress that is not expected to be attenuated in cytoplasmic envelopment. The ratio of eYFP-VP26⁺ pixels to gM-mCherry⁺ pixels at the JACs was not significantly different between the dfParental and ΔUS3 viruses (*Figure 5D*), suggesting that the lower ratio of eYFP-VP26⁺ pixels to gM-mCherry⁺ pixels in the JAC of ΔVP16-infected cells reflects poor capsid recruitment to these compartments in the absence of VP16. The eYFP-VP26⁺ pixels to gM-mCherry⁺ pixels ratio for the ΔpUL11, ΔpUL51, and ΔgE mutants was not significantly different from dfParental, suggesting capsid recruitment was unimpaired, and for ΔgK this ratio was significantly increased, suggesting that capsid recruitment to or retention at JACs is enhanced in cells infected with this mutant (*Figure 5D*).

Colocalisation analyses were performed to determine the overlap between capsids and gM-mCherry⁺ endomembranes at the JACs. Manders colocalisation coefficients, which are based on spatial coincidence, were chosen over intensity-based measures, such as Pearson's correlation coefficients, to minimize the influence of noise and cryoSIM reconstruction artefacts. Manders coefficient 1 (M₁) was calculated for each JAC and represented the number of eYFP-VP26⁺ pixels that overlapped with gM-mCherry⁺ pixels as a proportion of all eYFP-VP26⁺ pixels. Manders coefficient 2 (M₂) was also calculated for each JAC and represented the number of gM-mCherry⁺ pixels that overlapped with eYFP-VP26⁺ pixels as a proportion of all gM-mCherry⁺ pixels. These two measurements reflect the association of capsids with gM⁺ endomembranes. Both metrics were significantly lower for the ΔVP16 virus (N=22) compared with dfParental (N=16; *Figure 5E*, *Figure 5—figure supplement 1C*). The M₁ and M₂ coefficients for the mutants that formed small capsid clusters (i.e. ΔpUL11 [N=25] and ΔgE [N=23]) did not significantly differ from dfParental. For the mutants that form large capsid clusters, the M₁ coefficient was larger than dfParental for both ΔpUL51 (N=17) and ΔgK (N=18) but only reached statistical significance for ΔpUL51, and the M₂ coefficient was increased for both ΔgK and ΔpUL51 but only reached statistical significance for ΔgK. This suggests enhanced association of capsid and gM⁺



**Figure 5.** Cytoplasmic clustering of virus particles imaged using cryoSIM. U2OS cells were infected at MOI = 2 with indicated viruses for 16 hr. Cytoplasmic clustering of mutants was investigated using maximum Z projections of cryoSIM data. Scale bars = 10 μm. (**A**) gM-mCherry fluorescence was used to determine the cytoplasmic region in which virus assembly occurs, known as the juxtanuclear assembly compartment (JAC, dotted outline). The yellow corner markings denote the region of the dfParental virus-infected cell shown in **B**. (**B**) Cytoplasmic capsids at the JAC in ΔVP16-infected cells were less abundant and less closely associated with gM-mCherry⁺ endomembranes compared with those of dfParental-infected cells. (**C**) Representative images of JACs (dotted outline) for mutants. The JACs for the ΔUS3 virus were similar to those of the dfParental. ΔVP16 produced few cytoplasmic virus particles, ΔpUL11 and ΔgE produced small cytoplasmic clusters of capsids, and ΔpUL51 and ΔgK produced large cytoplasmic capsid clusters. The yellow corner markings denote the region of the ΔVP16 virus-infected cell shown in **B**. Note that the ΔUS3 example was reused in *Figure 4C*. (**D**) Thresholding of eYFP-VP26 and gM-mCherry fluorescence to filter out noise and background was performed and binary masks were

*Figure 5 continued on next page*

*Figure 5 continued*

produced. The ratio of eYFP-VP26⁺ pixels to gM-mCherry⁺ pixels at the JACs was reported. Data from the ΔUS3 virus were included as a negative control for attenuation in cytoplasmic virion assembly. (**E**) Manders coefficients ($M_1$) were measured for each virus and represent the amount of eYFP-VP26 fluorescence that colocalises with gM-mCherry (eYFP$_{coloc}$) as a proportion of all eYFP-VP26 fluorescence at the JAC (eYFP$_{total}$). $M_1$ values were markedly lower for the ΔVP16 virus than other viruses. $M_2$ values are shown in *Figure 5—figure supplement 1*. Mann-Whitney *U* tests were performed to assess significance of differences between dfParental (N=16) (green statistics) or ΔgK (N=18) (pink statistics) and other viruses, specifically ΔUS3 (N=9), ΔVP16 (N=22), ΔgE (N=23), ΔpUL11 (N=25), and ΔpUL51 (N=17). P-value thresholds: <0.05 (*), <0.005 (**), and <0.0005 (***). NS, no significance.

The online version of this article includes the following figure supplement(s) for figure 5:

**Figure supplement 1.** Association of gM-mCherry⁺ endomembranes with capsids.

endomembranes in the JAC for both these mutants but with different distributions of the components (*Figure 5E*).

CLXT imaging provided higher resolution details of the relative capsid and membrane arrangement for these mutants. As observed by cryoSIM (*Figure 5A*), cytoplasmic clusters of virus particles were not observed for dfParental HSV-1 by CLXT (*Figure 6A*). Small clusters of virus particles (<10 µm³) were observed in the JACs of the ΔpUL11 and ΔgE mutants (*Figure 6B*), and larger clusters of virus particles (≥10 µm³) were observed in the JACs of the ΔpUL51 and ΔgK viruses (*Figure 6C*). Additionally, linear arrays of capsids were observed in the cytoplasm of the ΔgK virus (*Figure 6D*) that were not observed in dfParental-infected cells (*Figure 3C*). CLXT revealed that these linear capsid arrays are not associated with gM-mCherry⁺ endomembranes (*Figure 6D*). From their linear organisation, we suspect they are located along filaments that could not be reliably resolved by cryoSXT, potentially microtubules, as these are known to be important for intracellular capsid transport (*Lee et al., 2006b*).

## 3D envelopment mechanism of HSV-1

The high penetrating power of X-rays enables the entire depth of the cell in each field of view to be imaged by cryoSXT, allowing rare or transient events to be captured (*Nahas et al., 2022a*). Combining this with the HSV-1 mutant strains that have impaired envelopment allows us to enrich intermediate stages of virus assembly that are otherwise extremely rapid and thus difficult to visualise. Numerous independent assembly intermediates of the ΔpUL51 virus were detected by CLXT (*Figure 7A–B* & *Figure 7—figure supplement 1A–B*). These included unenveloped capsids in the cytoplasm and capsids embedded in the surface of gM-mCherry⁺ vesicles. Note that the anterior and posterior faces of these vesicles cannot be reliably segmented owing to their lower contrast, a result of the 'missing wedge' in the tomographic data acquisition. Such vesicles were enriched in gM-mCherry at the pole near the capsids, indicating that gM-mCherry and potentially other viral proteins become concentrated at microdomains on vesicles prior to envelopment (*Figure 7A*). This observation is consistent with a previous hypothesis suggesting concentration of tegument and glycoproteins at the 'assembly pole' of the virion (*Maurer et al., 2008*). Virus particles budding into vesicles and fully enveloped virions were also observed (*Figure 7A–B*). Fully enveloped virions were distinguished from budding intermediates by observing differences in voxel intensity across the vesicles that contain them: the virions were separated from the vesicle membrane by a narrow volume of lumenal space that appeared brighter (*Figure 7—figure supplement 1B*), whereas budding events displayed a continuous drop in intensity between the membrane and the budding intermediate (*Figure 7—figure supplement 1A*). A 2D cross-section of a budding event (*Figure 7C–D*) appears topologically similar to envelopment events observed by 2D TEM (*Figure 7E*). This 3D view of cytoplasmic budding suggests envelopment is driven by capsid budding into spherical/ellipsoidal vesicles rather than by thin tubular endomembranes forming projections to wrap around capsids.

Capsid arrays were frequently observed near cytoplasmic vesicles for the ΔgK (*Figure 8A*), ΔpUL11 (*Figure 8B*), ΔpUL51 (*Figure 8C*), and ΔgE (*Figure 8D*) viruses. These features were not observed in dfParental-infected cells (*Figure 3C*), and they share four phenotypes regardless of the type of mutant. First, an array of capsids is located around one pole of the vesicle but not the antipole. Second, the capsids appear to be near the vesicles but generally do not appear embedded in the surface. Third, the pole of the vesicle near to the capsid arrays appears darker in the tomograms, indicating a greater presence of X-ray absorbing material. Fourth, CLXT revealed the vesicle pole near to the capsid arrays is enriched in gM-mCherry, suggesting gM and potentially other viral proteins accumulate at the pole nearer the capsid arrays, which could account for the greater abundance of X-ray absorbing material.



**Figure 6.** Correlative light X-ray tomography (CLXT) of cytoplasmic clusters of mutants. U2OS cells were infected at MOI = 2 with indicated viruses for 16 hr. (**A**) Individual envelopment events (arrows) as opposed to cytoplasmic clusters were observed for the dfParental virus. To aid visual comparison, this panel depicts the same data as shown in *Figure 3C*. C, cytoplasm; N, nucleus. (**B**) Cells infected with ΔpUL11 or ΔgE viruses contained small clusters of capsids (outlined) in the juxtanuclear assembly compartment (JACs). (**C**) Cells infected with ΔpUL51 or ΔgK viruses contained large clusters of capsids (outlined) in the JACs. (**D**) Linear arrays of eYFP-VP26⁺/gM-mCherry⁻ virus particles (outlined) were observed in the cytoplasm of a cell infected with ΔgK. Scale bars = 1 µm.

Based on these four shared phenotypes, we interpret these features to represent capsids interacting with appropriate target membranes but experiencing a delay or defect in their ability to bud into the vesicle lumen to acquire an envelope. To our knowledge, stalled envelopment events with these details have not been described by 2D TEM. The 3D nature of cryoSXT increases the likelihood that



**Figure 7.** 3D Snapshots of the trajectory of ΔpUL51 HSV-1 envelopment within the cytoplasm. U2OS cells were infected at MOI = 2 with ΔpUL51 HSV-1 for 16 hr. (**A**) Correlative light X-ray tomography (CLXT) revealed multiple stages in the assembly of the ΔpUL51 virus (arrows). (**B**) 3D renderings of the assembly stages captured by CLXT in A at different rotations around the X-axis. The vesicles appear open-ended because the anterior and posterior faces lack sufficient contrast for reliable segmentation — a result of the X-ray tilt series spanning 120° rather than 180°. (**C**) Capsids can be seen budding into vesicles. A cross-section through the middle gives the appearance of a capsid being wrapped by a tubular membrane. (**D**) A 2D cryo-soft-X-ray tomography (CryoSXT) projection of an envelopment event. (**E**) A 2D TEM of an envelopment event from an HFF-hTERT cell infected with an untagged wild-type (WT) HSV-1 (KOS strain). Scale bars = 1 μm.

The online version of this article includes the following figure supplement(s) for figure 7:

**Figure supplement 1.** Full and partial envelopment intermediates distinguished based on voxel intensity.



**Figure 8.** Mutants exhibit features of stalled envelopment. U2OS cells were infected at MOI = 2 with indicated viruses for 16 hr. (**A–D**) Arrays of unenveloped cytosolic capsids were observed near gM-mCherry-enriched vesicles for the indicated viruses. Vesicles were enriched in gM-mCherry at the pole nearest the capsid arrays. We interpret these features as capsids interacting with the appropriate target membranes but experiencing a delay or defect in budding. Some capsids appear to be located in the lumen of the vesicle (e.g. capsid marked with an asterisk (*) in **A**), but these are in fact external and located in front of or behind the open-ended segmentation of the vesicle. The vesicles appear open-ended because the X-ray tilt series spanned only 120°, causing the anterior and posterior faces to lack sufficient contrast for reliable segmentation. (**E–H**) For each indicated virus, the voxel intensity was measured at 30 points on the vesicles to provide a measure of the X-ray absorbing material present on the vesicle. Voxel intensities were plotted against the positions of the proximal capsids. Voxel intensities were measured from three projection planes spanning a depth of 30 nm and error bars show mean ± SD of these three planes. Two-tailed t-tests were performed to determine the significance of difference between the side of the vesicle nearest the capsids and the other side, as indicated by the dotted red line. Voxel intensity was lower on the side nearest the capsids, indicating this pole of the membrane contained a greater X-ray absorbing material. (**I**) A false-coloured heatmap of voxel intensity from the cryo-soft-X-ray tomography (cryoSXT) data was superimposed onto the vesicle segmentation from **D**. The vesicle is displayed at three angles with or without the

*Figure 8 continued on next page*

*Figure 8 continued*

proximal capsid arrays. The reciprocal of voxel intensity was used as a proxy for X-ray absorbance. The pole of the vesicle near the capsid arrays had greater X-ray absorption (red/purple) than the opposite pole (yellow/orange). Scale bars = 1 µm in **A–D** and 0.5 µm in **E–H**.

The online version of this article includes the following figure supplement(s) for figure 8:

**Figure supplement 1.** Features of gM-mCherry⁺ vesicles in infected cells.

**Figure supplement 2.** Impact of the gM-mCherry conjugation on the ΔgE virus.

ultrastructural features, such as these could be captured regardless of orientation with respect to the XY projection plane (*Harkiolaki et al., 2018*).

To quantitate the variation in X-ray absorbing material along the membrane from a stalled envelopment event for each virus, the voxel intensities of 30 points on the relevant vesicles were measured (*Figure 8E–H*). X-ray tomograms reconstructed by weighted back projection (WBP) were used for the analysis without applying any noise-averaging simultaneous iterations reconstruction technique (SIRT)-like filters (*Wolf et al., 2014*). Given that WBP tomograms are noisy (*Wolf et al., 2014*), voxel intensities were sampled from a 3×3 voxel matrix in the XY plane at each point on the vesicle and the minimum value was used. Voxel intensities were collected and averaged from three tandem tomographic projections spanning a depth of 30 nm. For all four mutants, the voxel intensity was lower on the vesicle pole nearer to the capsid arrays (*Figure 8E–H*). Voxel intensity is proportional to the X-ray radiation transmitted through the sample to the detector. A low voxel intensity suggests lower X-ray transmittance and greater X-ray absorption, indicating the vesicle pole nearer the capsid arrays contained a greater amount of carbon-rich X-ray-absorbing material. To visualise the variation in X-ray absorption around the vesicle from *Figure 8D* in 3D, the voxel intensity of the tomogram was superimposed and false-coloured onto a segmentation of the vesicle (*Figure 8I*). This illustrated that the vesicle pole nearer the capsid arrays had greater X-ray absorption than the antipole. The width of vesicles associated with capsid arrays were measured using the quantitation program *Contour* (*Nahas et al., 2022b*) and these vesicles had a mean width of 704.9±206.5 nm (mean ± SD, n=34) (*Figure 8—figure supplement 1A*). Stalled envelopment features were not observed in dfParental-infected cells, and we wondered whether the enrichment of gM-mCherry at one vesicle pole was an artefact of deleting tegument or envelope proteins that could directly or indirectly interact with gM (e.g. pUL11 or gE) (*Owen et al., 2015*). However, polarisation of vesicular gM-mCherry was observed in dfParental-infected cells in the absence of capsid arrays (*Figure 8—figure supplement 1B*). In this case, the vesicle appeared to be constricted between the gM-mCherry⁺ and gM-mCherry⁻ poles, which could arise for numerous reasons, such as fusion, fission, or pressure imposed by microtubules.

Stalled envelopment was frequently observed for the ΔgE virus (*Figure 8—figure supplement 2A*). This was surprising because no attenuation in replication kinetics was detected for the ΔgE virus when compared with dfParental (*Figure 2A*). Previous research indicated HSV-1 lacking both gE and gM was more severely attenuated in replication kinetics than HSV-1 lacking one of either protein (*Maringer et al., 2012*). This could be related to the indirect interaction between gE and gM via the VP22 tegument protein (*Figure 8—figure supplement 2B*; *Maringer et al., 2012*). Untagged and eYFP-VP26/gM-mCherry-tagged forms of the virus displayed similar replication kinetics, suggesting no detectable impact on replication of mCherry conjugation to the C terminus of gM (*Figure 8—figure supplement 2C*). Plaque sizes were measured for each virus to determine if the tagging affected cell-to-cell spread. The ΔgE virus was on average 0.65x smaller than the WT virus for the untagged form but 0.35x smaller for the tagged form, suggesting the tagging contributed to an attenuation in cell-to-cell spread (*Figure 8—figure supplement 2D–E*).

## Width measurements of virus assembly intermediates

Electron cryo-tomography (cryoET) has been used to measure the width of the tegument layer and an asymmetric distribution of tegument around the capsid has been demonstrated, with widths of 5 nm and 35 nm at opposing poles (*Grünewald et al., 2003*). Widths of viral assembly intermediates were measured for the dfParental, ΔpUL51, and ΔgE viruses by cryoSXT to determine if the added width of the tegument layer around the capsid could be detected (*Figure 9* & *Table 1*; *Nahas et al., 2022a*). No difference was observed in the width of nuclear capsids and unenveloped membrane-proximal capsids by cryoSXT (*Figure 9* & *Table 1*). This could indicate that recruitment of tegument to capsids

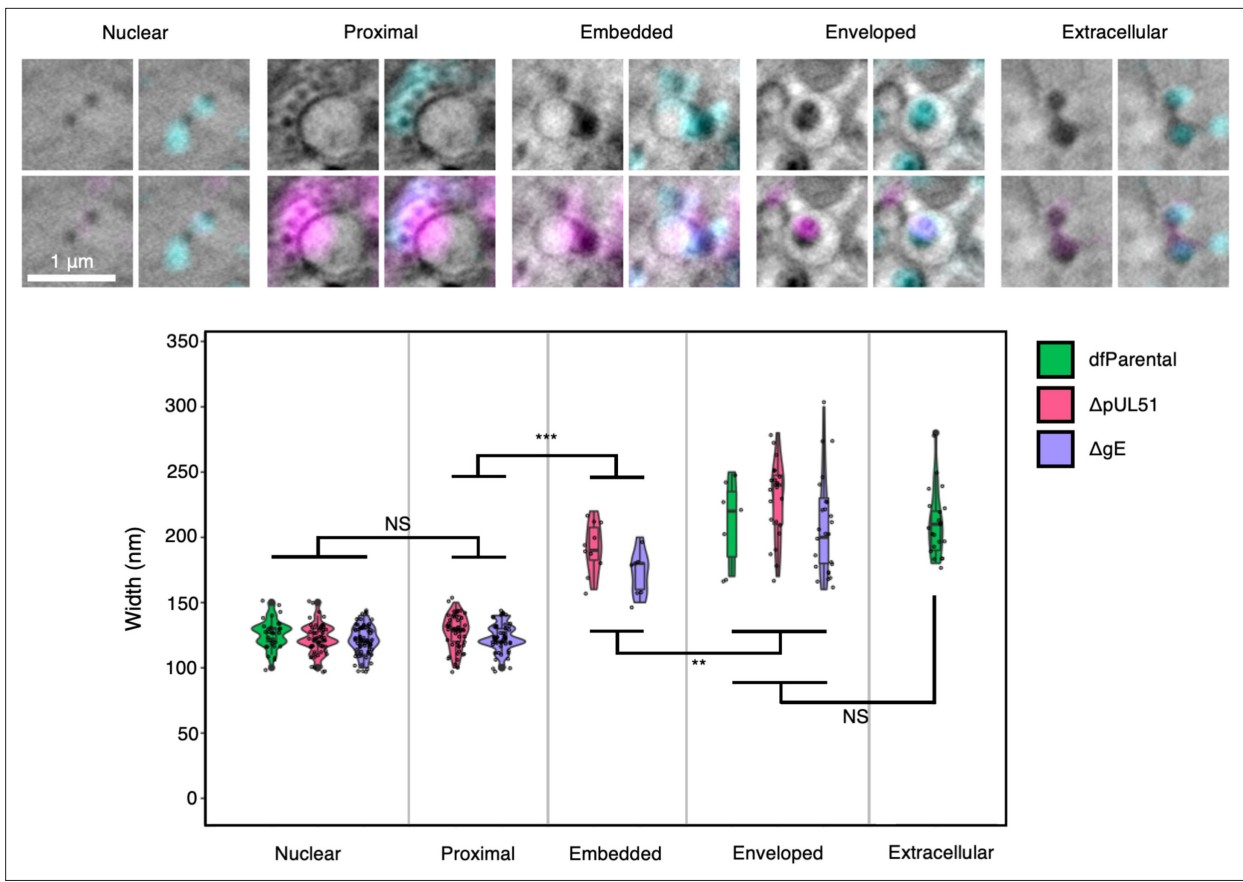

**Figure 9.** Cryo-soft-X-ray tomography (CryoSXT) resolved differences in the widths of virus assembly intermediates. U2OS cells were infected at MOI = 2 for 16 hr. Widths of virus particles at different stages of assembly were measured for the dfParental, ΔpUL51, and ΔgE viruses. See *Table 1* for values. Mann-Whitney *U* tests were used to assess the significance of differences. No significant differences in width were observed between virus musctants at the same stage of assembly, except for enveloped ΔgE (N=25) and ΔpUL51 (N=22) virus assembly intermediates (*p*-value = 0.0285). P-value thresholds: <0.05 (*), <0.005 (**), and <0.0005 (***). NS, no significance. Scale bars = 1 μm.

does not occur until cytoplasmic envelopment. However, ICP0, ICP4, pUL36, pUL37, and pUS3 are thought to condense on capsids in the nucleus, demonstrating that tegument condensation begins early in assembly (*Henaff et al., 2013*; *Bucks et al., 2007*). Alternatively, it is possible that the tegument layer is too diffuse around cytoplasmic capsids such that it does not produce detectable X-ray absorption by cryoSXT. Under this scenario, X-ray absorption may become detectable if the tegument layer were to compress around capsids upon membrane-embedding and budding. Embedded particle widths were measured normal (at 90°) to the membrane to limit distortion of the measurements by the membrane. The widths of embedded particles (184.12±4.86 nm SEM; N=17; range

**Table 1.** Width of virus assembly intermediates measured using cryo-soft-X-ray tomography (cryoSXT). U2OS cells were infected at MOI = 2 for 16 hr. Data are combined from dfParental, ΔpUL51, and ΔgE viruses. See *Figure 9* for a graphical representation. SD, standard deviation; SEM, standard error of the mean; N, number of virus assembly intermediates used for analysis.

|  | Mean (nm) | SD (nm) | SEM (nm) | N |
|---|---|---|---|---|
| Nuclear | 122.5 | 11.4 | 0.9 | 149 |
| Proximal | 125.0 | 11.8 | 1.1 | 105 |
| Embedded | 184.1 | 20.0 | 4.9 | 17 |
| Enveloped | 216.9 | 34.2 | 4.7 | 54 |
| Extracellular | 210.0 | 25.9 | 5.6 | 21 |

150–220 nm; SD 20.02 nm) were significantly greater than nuclear capsids (122.48±0.94 nm SEM; N=149; range 100–150 nm; SD 11.44 nm; Mann-Whitney *U*-test *p*-value = 4.27 × 10$^{-12}$) or membrane-proximal capsids (124.95±1.15 nm SEM; N=105; range 100–150 nm; SD 11.78 nm; Mann-Whitney *U*-test *p*-value = 1.79 × 10$^{-11}$), suggesting the tegument layer condenses and compresses around the capsid upon membrane-embedding and budding (*Figure 9* & *Table 1*). The widths of embedded particles were also significantly lower than intracellular enveloped particles (216.85±4.65 nm SEM; N=54; range 160–300 nm; SD 34.19 nm; Mann-Whitney *U*-test *p*-value = 6.8 × 10$^{-4}$) and extracellular particles (210.00±5.65 nm SEM; N=21; range 180–280 nm; SD 25.88 nm; Mann-Whitney *U*-test *p*-value = 3.5 × 10$^{-3}$), consistent with incomplete membrane acquisition.

## Discussion

Application of correlative cryoSIM and cryoSXT imaging to the study of viral infection has thus far been limited to visualising the entry of reovirus during infection and has not yet been used to study other stages of virus assembly, nor has it been used to study phenotypes of mutant viruses (*Kounatidis et al., 2020*). In this study, nine mutants of HSV-1, each lacking a protein involved in virus assembly, were generated to study attenuation in nuclear egress and cytoplasmic envelopment with correlative cryoSIM and cryoSXT imaging. Each mutant genetically encoded fluorescent capsid (eYFP-VP26) and envelope (gM-mCherry) proteins that allowed identification of viral assembly intermediates without the need for chemical fixation, cell permeabilization or immunostaining, all of which could introduce ultrastructural artefacts. Instead, samples were vitrified in a physiologically relevant near-native state by plunge-cryocooling (*Harkiolaki et al., 2018*). Lack of protein expression for each mutant gene was verified (*Figure 1C–D*); the replication kinetics and plaque sizes for each virus were assessed (*Figure 2A–C*); roles of pUL16, pUL21, pUL34, VP16, and pUS3 in nuclear egress of HSV-1 were observed (*Figure 4*); CLXT revealed different phenotypes associated with attenuation for cytoplasmic envelopment with the ΔpUL11, ΔVP16, ΔpUL51, ΔgK, and ΔgE viruses, suggesting the corresponding proteins possess different functions in envelopment (*Figures 5–8*); and the widths of virus assembly intermediates identified by CLXT were measured (*Figure 9* & *Table 1*). The results of our CLXT analysis for each HSV-1 mutant are summarised in *Table 2*.

The large defect in nuclear egress of HSV-1 lacking pUL34, as assessed by cryoSIM and CLXT observation (*Figure 4B–D*), is consistent with its role as an essential component of the viral NEC. A nuclear egress defect was apparent but was significantly less pronounced for the ΔUS3 mutant, consistent with the impact of pUS3 kinase-regulated NEC activity and nuclear lamina dispersal being variable in magnitude and cell-type specific (*Thorsen et al., 2021*; *Mou et al., 2009*; *Masoud Bahnamiri and Roller, 2021*; *Bjerke and Roller, 2006*). While both pUL16 and pUL21 have been shown to promote nuclear egress in HSV-2 (*Le Sage et al., 2013*; *Gao et al., 2017*), previous studies have not confirmed a role for either protein in nuclear egress in HSV-1 (*Sarfo et al., 2017*; *Gao et al., 2018*). Our cryoSIM and CLXT data (*Figure 4B–D*) show that both pUL16 and pUL21 promote nuclear egress in U2OS cells. Of note, the ΔpUL21 mutant was found to contain a non-synonymous point mutation in pUS3 that may independently influence nuclear egress (*Figure 2—figure supplement 2*), consistent with our previous data showing positive selection of pUS3 mutations during passage of pUL21 mutant viruses that rescues HSV-1 replication and spread (*Benedyk et al., 2021*). While the presence of the pUS3 C463T mutation in our eYFP-VP26 and gM-mCherry tagged ΔpUL21 virus may be expected to compensate for any defect in nuclear egress in order to restore viral fitness, the nuclear egress defect observed for the ΔpUL21 mutant is greater than for the ΔUS3 mutant (*Figure 4B*). This suggests that, if anything, our cryoSIM and CLXT data may underestimate the importance of pUL21 in this process. In addition, we note that our CLXT studies cannot differentiate mature genome-containing 'C' capsids from the genome-free 'A' and 'B' capsids that are known to accumulate in the absence of HSV-1 pUL16 (*Gao et al., 2018*) and pUL21 (*Sarfo et al., 2017*; *Thomas et al., 2022*). The reduced extent of nuclear egress at 16 hpi for the ΔVP16 virus (*Figure 4B–D*) could be attributed to a slower rate of replication in the absence of VP16 (either via diminished immediate-early gene trans-activation *Campbell et al., 1984* or via loss of regulation of the HSV-1 viral host shutoff protein *Knez et al., 2003*) rather than an attenuation in nuclear egress. However, the results are consistent with previous research that showed VP16 is incorporated on perinuclear virus particles in HSV-1 (*Naldinho-Souto et al., 2006*) and that cells infected with a ΔVP16 virus produce clusters of virus particles in the perinuclear space (*Mossman*

**Table 2.** Summary of findings for each HSV-1 gene.

| Gene | Protein | Observations | Interpretation |
|---|---|---|---|
| UL11 | pUL11 | • 10–100-fold reduction in replication observed in single-step virus growth assays<br>• Small plaques in plaque assays, similar in size to those of ΔgE mutant<br>• Small capsid clusters observed at the JAC using cryoSIM<br>• Small capsid clusters (<10 µm$^3$) observed at the JAC using CLXT, with capsid arrays near vesicle pole enriched in gM-mCherry | • Minor role in virus assembly affecting cytoplasmic envelopment and cell-cell spread<br>• Involved in the budding stage of cytoplasmic envelopment<br>• Similar importance in cell-cell spread as binding partner gE |
| UL16 | pUL16 | • 10–100-fold reduction in replication observed single-step virus growth assays, similar to ΔpUL11 and ΔpUL21 mutants<br>• Small plaques, but larger than those of the ΔpUL11 mutants<br>• High N:C capsid ratio | • Required for efficient nuclear egress but less important than the essential protein pUL34<br>• Unable to assess involvement in cytoplasmic envelopment due to paucity of cytoplasmic capsids |
| UL21 | pUL21 | • 10–100-fold reduction in replication observed single-step virus growth assays.<br>• High N:C capsid ratio | • Required for efficient nuclear egress but less important than the essential protein pUL34<br>• Unable to assess involvement in cytoplasmic envelopment due to paucity of cytoplasmic capsids |
| UL34 | pUL34 | • 10$^5$-fold reduction in replication observed single-step virus growth assays<br>• No plaques in plaque assay<br>• Highest N:C capsid ratio | • Major/essential role in nuclear egress |
| UL48 | VP16 | • No replication detected at 24 hr post-infection in single-step virus growth assays<br>• Very small plaques in 72 hr plaque assay<br>• High N:C capsid ratio but significantly lower than ΔUL34<br>• No capsid clustering at JAC<br>• Capsids associated less frequently with gM-mCherry$^+$ endomembranes | • Major/essential role in virus assembly<br>• Required for efficient nuclear egress but less important than pUL16, pUL21, and pUL34<br>• Involved in the recruitment of capsids to endomembranes during cytoplasmic envelopment |
| UL51 | pUL51 | • 10–500-fold reduction in replication observed in single-step virus growth assays<br>• Very small plaques in plaque assay<br>• Large capsid clusters observed at the JAC using cryoSIM<br>• Large capsid clusters (≥10 µm$^3$) observed at the JAC using CLXT, with capsid arrays observed near vesicle pole enriched in gM-mCherry | • Major role in the budding stage of cytoplasmic envelopment<br>• Unable to assess involvement in cell-cell spread due to major attenuation in cytoplasmic envelopment |
| UL53 | gK | • 10$^3$–10$^5$-fold reduction in replication observed in single-step virus growth assays<br>• No plaques in plaque assay<br>• Large capsid clusters observed at the JAC using cryoSIM<br>• Large capsid clusters (≥10 µm$^3$) observed at the JAC using CLXT, with capsid arrays observed near vesicle pole enriched in gM-mCherry<br>• Linear arrays of capsids not associated with gM-mCherry$^+$ endomembranes observed using CLXT | • Major/essential role in the budding stage of cytoplasmic envelopment<br>• Cytoplasm becomes saturated with capsids in the absence of this protein, potentially forcing capsids approaching the JAC to stall on microtubules |
| US3 | pUS3 | • 10-fold reduction in replication observed in single-step virus growth assays<br>• Largest plaques of all mutant viruses<br>• Lowest N:C ratio of all mutants involved in nuclear egress | • Minor role in nuclear egress<br>• Only minor contributions to overall virus assembly and spread |
| US8 | gE | • No defects detected in single-step virus growth assays<br>• Small plaques in plaque assays, similar in size to those of ΔpUL11 mutant<br>• Small capsid clusters observed at the JAC using cryoSIM<br>• Small capsid clusters (<10 µm$^3$) observed at the JAC using CLXT, with capsid arrays observed near the vesicle pole enriched in gM-mCherry | • Not required for virus assembly, but does contribute as delayed envelopment was observed by CLXT<br>• Major role in cell-cell spread |

*et al., 2000*). The significant defect or delay in nuclear egress observed in this study (*Figure 4B–D*) indicates that VP16 could play a larger role in nuclear egress than previously supposed.

In addition to nuclear egress, the CLXT workflow presented here represents a powerful platform for comparative investigation of viral cytoplasmic envelopment. Substantial cytoplasmic envelopment defects were observed for the ΔpUL11, ΔpUL51, ΔgE, ΔgK, and ΔVP16 mutants (*Figures 5 and 6*), but the precise phenotype of the defect differed between mutants. Capsids were largely absent from juxtanuclear gM-mCherry⁺ endomembranes in cells infected with the ΔVP16 virus. This is consistent with the highly abundant tegument protein VP16 playing a key role in bridging inner tegument proteins with the outer tegument and viral glycoproteins during assembly, as well as the severe replication deficiencies of this mutant at 24 hpi (*Figure 2A*; *Svobodova et al., 2012*; *Ko et al., 2010*; *Elliott et al., 1995*; *Gross et al., 2003*; *Kamen et al., 2005*; *Kato et al., 2000*; *Smibert et al., 1994*; *Schmelter et al., 1996*).

Cells infected with ΔpUL11 and ΔgE HSV-1 had small clusters of capsids in JACs (*Figures 5 and 6*), consistent with previous TEM studies (*Baines and Roizman, 1992*; *Fulmer et al., 2007*; *Farnsworth et al., 2003*; *Farnsworth and Johnson, 2006*; *Johnson et al., 2001*), and both had moderate defects in virus replication and/or spread (*Figure 2A–C*), suggesting that both play modest roles in cytoplasmic envelopment. Removal of pUL51 or gK gave rise to much larger clusters of capsids (*Figures 5 and 6*) and more extensive replication and spread defects (*Figure 2A–C*), suggesting more severe deficiencies in capsid envelopment. pUL51 and its binding partner pUL7 have established roles in stimulating the cytoplasmic wrapping of nascent virions (*Albecka et al., 2017*; *Roller and Fetters, 2015*), and the 3D structure of pUL51 closely resembles components of the cellular endosomal sorting complex required for transport (ESCRT)-III machinery (*Butt et al., 2020*), but the precise molecular roles pUL7 and pUL51 play in capsid envelopment remain unclear. Two replication curves reveal that removal of pUL51 expression causes a wide-ranging 10- to 500-fold reduction in virus replication (*Figure 2A*, *Figure 2—figure supplement 1*), which is consistent with data showing that inhibition of ESCRT activity via expression of a dominant negative form of the ESCRT-associated cellular ATPase VPS4A effectively prevents HSV-1 budding (*Crump et al., 2007*). CLXT reveals large clusters of cytoplasmic virus particles and stalled envelopment events for ΔpUL51 HSV-1 (*Figures 5 and 6*), suggesting that pUL51 acts as a catalyst to accelerate cytoplasmic envelopment but that additional (redundant) cellular and/or viral mechanisms support envelopment in the absence of pUL51.

Disruption of the pUL20-gK complex causes substantial defects in cytoplasmic envelopment and the cytoplasm of cells infected with HSV-1 ΔgK or ΔUL20 mutants has been shown to harbour a greater number of unenveloped membrane-associated capsids (*Fulmer et al., 2007*; *Lau and Crump, 2015*). The gK/pUL20 complex has been shown to interact with pUL37, and TEM analysis revealed that unenveloped capsids accumulate in the cytoplasm if the interacting residues in pUL37 are mutated (*Chouljenko et al., 2016*; *Jambunathan et al., 2014*), suggesting that interactions between gK/pUL20 and pUL37 are required for the formation of virus assembly compartments. However, our study revealed that capsids still associate with membranes in the absence of gK, indicating an interaction of gK/pUL20 with pUL37 is not required for recruiting capsids to assembly compartment membranes (*Figures 5 and 6*).

What our CLXT data does demonstrate is that the activity of both gK and pUL51 (and by association the gK/pUL20 and pUL7/pUL51 complexes) are important for virion assembly to progress beyond association of capsids with cytoplasmic membranes. Whether both these complexes function to regulate the similar cellular machinery involved in membrane curvature and/or scission events that are required to complete the cytoplasmic envelopment process, for example, ESCRT activity, remains to be established.

In addition to extensive membrane-associated cytoplasmic capsids being observed in ΔgK-infected cells, linear arrays of unenveloped capsids were observed in the cytoplasm for this virus (*Figure 6D*). Cytoplasmic capsids migrate along microtubules and these linear arrays could represent capsids stalled on microtubules when envelopment compartments become saturated with capsids in the absence of gK (*Lee et al., 2006b*).

One surprising result was the observation of numerous stalled envelopment events in ΔgE infected cells, despite no defects in replication kinetics being observed for the ΔgE virus (*Figure 2A*, *Figure 8—figure supplement 2C*). This could arise from subtle differences in the kinetics of envelopment, whereby a minor defect in ΔgE budding slows the rapid process of envelopment sufficiently

for stalled budding profiles to be observed, but not for long enough to cause a significant defect in virus replication.

Previous ultrastructural analysis of HSV-1 has largely been performed using TEM, which retains some advantages over cryoSXT. TEM offers higher resolution, allowing different components of the virus assembly intermediates to be unambiguously identified, such as the capsid, tegument, and envelope, thereby negating the need for fluorescence correlation with genomically-encoded tags. This allows HSV strains lacking fluorescent fusion proteins to be imaged, reducing the risk that the fluorescent proteins attached to structural proteins could interfere with virus assembly. However, 2D cross-sections of cytoplasmic envelopment events visualised by TEM, which appear as a capsid encircled by a tubular C-shaped endomembrane, are compatible with more than one 3D model of envelopment; techniques, such as cryoSXT that allow greater volumes of the cell to be imaged than TEM are required to clarify the envelopment mechanism. Our 3D CryoSXT analysis reveals numerous budding events occurring in spherical/ellipsoidal vesicles and fully enveloped virions within spherical/ellipsoidal carrier vesicles (*Figure 7*). We, therefore, propose a model for HSV-1 envelopment wherein the apparently C-shaped endomembranes observed in TEM represent the cross-sectional appearance of a spherical or ellipsoidal vesicle deformed by budding of a capsid into the vesicle lumen (*Figure 7*). Recycling endosomes and *trans*-Golgi network vesicles have been identified as potential envelopment organelles, so the vesicles observed in this study may represent these compartments or a fusion hybrid (*Ahmad and Wilson, 2020*; *Hollinshead et al., 2012*). Fluorescently tagged markers of vesicular compartments could aid with determining the nature of the envelopment organelle using CLXT in future work.

We observed several instances of polarised arrays of capsids around spherical vesicles for mutant viruses with delayed envelopment (ΔpUL11, ΔpUL51, ΔgK, and ΔgE) (*Figure 8*). To the best of our knowledge, such observations have not been made in TEM studies of HSV-1 assembly with wild-type or mutant viruses. This discrepancy may result from the low probability of capturing multiple capsids bordering a vesicle within a single TEM ultrathin section. Our cryoSXT imaging shows a greater density of material on the side of the vesicle membrane nearer the capsid (*Figure 8*), a feature that has also been observed in TEM (*Hollinshead et al., 2012*). One advantage of our CLXT imaging is the ability to extend such insights by revealing the enrichment of specific viral proteins (exemplified in our data by gM-mCherry) at the capsid-proximal pole of the vesicle. The observations are consistent with the hypothesis that HSV-1 budding is asymmetric, being initiated at a 'budding pole' that is rich in tegument and glycoproteins (*Maurer et al., 2008*). In the future, probes for different viral and host membrane proteins and specific lipid species in CLXT could shed light on membrane partitioning processes that occur during herpesvirus assembly.

An advantage of the relatively high-throughput analysis afforded by cryoSXT (compared to TEM and cryoET) is the ability to obtain robust particle size data *in cellulo*. No difference was observed in the measured width of nuclear capsids compared with membrane-proximal cytoplasmic capsids (*Figure 9* & *Table 1*), suggesting that there is only minimal recruitment of tegument around cytosolic capsids and/or that the tegument is too diffuse to produce detectable X-ray absorption. However, the average width of envelopment intermediates for ΔgE and ΔpUL51, where capsids are engaged with the membrane ('embedded'), is only approximately 32.7 nm lower than that of fully enveloped extracellular enveloped virions (*Figure 9* & *Table 1*). This suggests that the tegument layer becomes compressed around capsids immediately prior to, or concomitantly with, virus budding.

In conclusion, our multi-modal imaging strategy has provided novel ultrastructural insight into HSV-1 assembly, allowing the assembly trajectory of wild-type and mutant viruses to be observed in 3D. This revealed that envelopment occurs by the lumenal budding of capsids at spherical/ellipsoidal vesicles, rather than by wrapping of tubular membranes around capsids, and that tegument compression is concomitant with budding. Polarised arrays of capsids at cytoplasmic vesicles were observed for several mutants, suggesting envelopment is focused to one side of the endomembrane, and CLXT imaging suggested that these capsid-proximal surfaces are enriched in viral glycoproteins. Previously uncharacterised defects in nuclear egress were observed for HSV-1 lacking VP16, pUL16, and pUL21. Furthermore, comparative analysis reveals that deletion of VP16, pUL11, gE, pUL51, or gK cause distinct defects in cytoplasmic envelopment. Our data highlight the contributions that key HSV-1 envelope and tegument proteins make to virus assembly and underscore the power of correlative fluorescence and X-ray tomography cryo-imaging for interrogating virus assembly.

## Materials and methods

### Reagents

Quantifoil 3 mm gold TEM grids with a holey carbon film (R 2/2, 200 mesh) were used as a substrate for cells prepared for cryopreservation. TEM grids were treated with poly-L-lysine (Sigma Aldrich). 150 nm gold fiducials were used to align cryoSXT projections (Creative Diagnostics Nanoparticle AF647).

### Cell lines

U2OS cells (ATCC Cat# HTB-96; RRID:CVCL_0042) and African green monkey kidney (Vero) cells (ATCC Cat# CRL-1586, RRID:CVCL_0574) were cultured in Dulbecco's Modified Eagle's Medium (DMEM; Thermo Fisher Scientific) containing 10% (v/v) fetal bovine serum (FBS; Capricorn), 4 mM L-glutamine (Thermo Fisher Scientific), and penicillin/streptomycin (10000 U/mL; Thermo Fisher Scientific). Hanks' Balanced Salt Solution (HBSS; Thermo Fisher Scientific) and 0.25% Trypsin-EDTA (Thermo Fisher Scientific) were used to wash and detach adherent cells, respectively. Cells were maintained in a humidified 5% $CO_2$ atmosphere at 37°C. All cell lines were regularly tested and confirmed to be mycoplasma-negative. The cell lines were not STR-profiled because they have been carefully managed to avoid any chances of cross-contamination since cultures were received from ATCC and confirmation of the identity of the cell lines is not pertinent to this study.

### Generation of HSV-1 mutants

Mutants were generated from a bacterial artificial chromosome carrying the KOS HSV strain (*Gierasch et al., 2006*). A mutant was previously reconstituted from this system containing eYFP-VP26 and gM-mCherry (*Scherer et al., 2021*). This was used as the dfParental virus and the BAC from which it was derived was used as a template for the mutagenesis of other mutants. A two-step red recombination system (*Tischer et al., 2010*) was used to generate the mutants with the primers shown in *Supplementary file 1A*. The ΔpUL11 virus was generated by mutating the start codon and introducing one stop codon at residue 3 of the UL11 open reading frame (ORF). The ΔpUL16 virus was generated by introducing three stop codons at residue 15 of the UL16 ORF. The ΔpUL21 virus was generated by introducing three stop codons at residue 23 of the UL21 ORF. The ΔUL34 virus was generated by deleting codons 1–228 from the UL34 ORF. The ΔVP16 virus was generated by deleting codons 1–478 from the UL48 ORF. The ΔpUL51 virus was generated by introducing three stop codons at residue 21 of the UL51 ORF. The ΔgK virus was generated by introducing three stop codons at residue 54 of the UL53 ORF. The ΔUS3 virus was generated by deletion of the entire coding sequence of US3. The ΔgE virus was generated by introducing three stop codons at residue 21 of the US8 ORF. Mutant BACs were transfected, together with a Cre recombinase expression plasmid (pGS403) to excise the BAC cassette from the KOS genome, into 6-well plates containing 70–80% confluent Vero cells to reconstitute the viruses, other than ΔUL34, ΔVP16, and ΔgK, which were transfected into the respective complementing Vero-modified cell lines: UL34CX, 16_8, and VK302 (*Hutchinson and Johnson, 1995*; *Roller et al., 2000*; *Weinheimer et al., 1992*). Successive stocks were generated by infecting Vero cells or complementing cells at MOI = 0.01. Cells were harvested once all cells demonstrated cytopathic effect, freeze-thawed, and sonicated at 50% amplitude for 40 s in a cuphorn sonicator. Final stocks were clarified by centrifugation at 3200×g for 5 min in a benchtop centrifuge, divided into 10–20 μL aliquots and were stored at –70°C. All virus stocks were titrated on U2OS cells, Vero cells, or Vero-modified complementing cells.

### Infection assays

For the immunoblots, Vero cells (*Figure 1C–D*) were seeded in 6-well plates, were left to reach 70–80% confluency, were infected at MOI = 5 with indicated viruses in a minimal volume of medium (500 μL) and were incubated in a 5% $CO_2$, 37°C incubator. After 1 hr, the inoculum was diluted to 2 mL with fresh medium and cells were incubated overnight.

For the single-step replication curves (*Figure 2A*, *Figure 2—figure supplement 1*, and *Figure 8—figure supplement 2C*), U2OS cells were seeded in 24-well plates at a density of 1×10^5 cells per well and were infected the next day at MOI = 2 with the indicated viruses in a minimal volume of medium (250 μL) for 1 hr in a 5% $CO_2$, 37°C incubator. After 1 hr, the inoculum was aspirated off the cells and the cells were treated with citric acid solution (40 mM citric acid pH 3, 135 mM NaCl, and 10 mM KCl)

for 1 min to inactivate unabsorbed virus. Citric acid solution was subsequently aspirated off the cells and the samples were washed thrice with 500 µL PBS before adding 500 µL fresh medium.

For the plaque size assays (*Figure 2B–C* and *Figure 8—figure supplement 2D–E*), U2OS cells were seeded on 6-well plates, were left to reach ~90% confluency, and were infected with a low titer of the indicated viruses (calculated to produce an average of 30 plaques per well). Cells were incubated with reduced serum (2% v/v) medium supplemented with 0.3% high viscosity carboxymethyl cellulose and 0.3% low viscosity carboxymethyl cellulose for 72 hr.

For cryoSIM and cryoSXT experiments (*Figures 3–8*, *Figure 5—figure supplement 1*, *Figure 7—figure supplement 1* and *Figure 8—figure supplement 1*), TEM grids were treated by glow discharge and were incubated in filtered poly-L-lysine for 10 min in 6-well plates as described previously (*Nahas et al., 2022a*; *Okolo et al., 2021*). Poly-L-lysine was aspirated off and U2OS cells were seeded onto the holey-carbon-coated side of the grids at a density of $3×10^5$ cells per well. After overnight culture in a 5% $CO_2$ and 37°C incubator, the cells were infected at MOI = 2 with indicated viruses in a minimal volume of medium (500 µL) and were incubated in a 5% $CO_2$, 37°C incubator. After 1 hr, the inoculum was diluted to 2 mL with fresh medium and were incubated for an additional 14.5 hr. Samples were washed twice with 1 mL serum-free medium and were then overlain with a staining solution containing 50 nM MitoTracker Deep Red (Thermo Fisher Scientific) in serum-free medium. For samples infected with the dfParental, ΔpUL16, ΔpUL21, ΔUL34, ΔVP16, and ΔUS3 viruses, the staining solution also contained 2 µg/mL Hoescht33342 (Thermo Fisher Scientific). Samples were washed twice with serum-free medium after 30 min of staining and the grids were loaded into a Leica EM GP2 plunge freezer set to 80% humidity. A working solution of gold fiducials was prepared by centrifugation of a 1 mL stock (provided in the *Reagents* section) at 12×g for 5 min at RT. The pellet was resuspended in HBSS and the working solution was sonicated at 80 kHz (100% power) and 6°C to disperse clumps of gold fiducials. A 2 µL solution of the working solution was overlain onto the holey-carbon coated side of the grids, which were then blotted for 0.5–1 s on the opposite side using Whatman paper. Grids were then immediately plunged into liquid nitrogen-cooled liquid ethane and were transferred into storage containers maintained under liquid nitrogen.

For immunoblots, infected cells were washed twice with 1 mL PBS, scraped off 6-well plates, and were centrifuged at 2000×g for 5 min. Pellets were isolated and lysed with a solution of Complete Protease Inhibitor without EDTA (Roche) diluted 1 in 10 in a lysis buffer (Sigma Aldrich) on ice for 20 min Insoluble material was removed by centrifugation at 20,000×g, 4°C for 10 min. Supernatants were resuspended in SDS-PAGE loading buffer supplemented with 2-mercaptoethanol. To immunoblot pUL20 and gE, the mixtures were heated to 42°C for 20 min. To immunoblot the other proteins, separate mixtures were boiled in a water bath for 5 min. Samples were resolved on SDS-PAGE gels alongside a Blue Protein Standard Broad Range ladder (New England BioLabs). Bands were transferred onto nitrocellulose membranes and were blocked with a 5% (w/v) solution of milk powder in PBS. The following primary antibodies were used: anti-pUL11 (*Leege et al., 2009*) at 1:1000, anti-pUL16 (*Carmichael and Wills, 2019*) at 1:1000, anti-pUL21 (*Benedyk et al., 2021*) at 1:1, anti-pUL34 (*Reynolds et al., 2002*) at 1:500, anti-VP16 (*McLean et al., 1982*) at 1:10, anti-pUL51 (3D3) (*Albecka et al., 2017*) at 1:1, anti-pUL20 (*Lau and Crump, 2015*) at 1:1000, anti-pUS3 (*Finnen et al., 2010*) at 1:1000, anti-gE (*Cross et al., 1987*) at 1:10, anti-VP5 (*McClelland et al., 2002*) at 1:10, and anti-GAPDH at 1:1000 (GeneTex Cat# GTX28245, RRID:AB_370675). Membranes were stained with the following secondary antibodies: IRDye 680T conjugated goat anti-rat (926–68029), donkey anti-rabbit (926–68023) or goat anti-mouse (926–68020), or LI-COR IRDye 800CW conjugated donkey anti-rabbit (926–32213), donkey anti-chicken (926–32218), or goat anti-mouse (926–32210). Primary and secondary antibody solutions were generated in PBS-T supplemented with 0.5% milk powder, membranes were washed four times for 5 min on a rocking platform with PBS-T after each round of antibody staining. For membranes immunoblotted for pUL16 or pUL21, TBS, and TBS-T were used instead of PBS and PBS-T. An Odyssey CLx Imaging System (LI-COR) and Image Studio Lite Software (LI-COR) were used to visualise immunoblots.

## Single-step replication curves

Infected samples were prepared as described in *Infection Assays* and were transferred to –70°C storage at 2, 6, 9, 12, and 24 hr post infection. After at least 2 hr of storage at –70°C, the samples were thawed at 37°C. This process of freeze-thawing was repeated and the thawed samples were

scraped off the 24-well plates using the blunt end of a 1 mL plunger and were transferred into 1.5 mL microcentrifuge tubes for –70°C storage for at least 2 hr. 10-fold serial dilutions of the samples were performed and titrated on Vero cells as previously described (*Albecka et al., 2017*). To logarithmically transform the PFU data, 0 values on the exponential scale were converted to 1.

## Sequencing and alignment

Working stock solutions of ΔgK, ΔpUL21, and dfParental HSV-1 were prepared for Sanger sequencing by adding 10 µL of virus stock to 200 µL of a Quantilyse solution (*Pierce et al., 2002*). Samples were heated to 55°C in a thermal cycler for 2.5 hr, 85°C for 45 min, and were left on hold at 10°C. 2 µL of these solutions were used as templates for PCR amplification with custom primers (*Supplementary file 1B*). Amplicon products were purified (EconoSpin, Epoch Life Science) and analysed by Sanger Sequencing with custom sequencing primers (*Supplementary file 1C*). The HSV-1 strain KOS UL53 sequence (encoding gK) was aligned with the sequencing results using Clustal Omega and were visualised using Jalview (*Macdonald et al., 2012*; *Sievers and Higgins, 2014*; *Waterhouse et al., 2009*). Tracking of indels by decomposition (TIDE) analysis was performed to identify prevalent base pair changes in the US3 gene in the ΔpUL21 mutant (*Brinkman et al., 2014*).

## Plaque size assays

Infected cells were prepared as described in *Infection Assays* and were washed with 1 mL PBS at 72 hpi. Samples were fixed with 1 mL 4% formaldehyde for 10 min and were washed with 1 mL PBS. Samples were blocked for 30 min with a solution of 5% FBS and 0.1% Tween-20 in PBS on a rocking platform. Samples were incubated for 1 hr with a 1:10 dilution of anti-gD (LP2) in blocking buffer on a rocking platform (*Minson et al., 1986*). Samples were washed with blocking buffer 3 times and were incubated for 1 hr with a goat-derived anti-mouse IgG (H+L) secondary antibody conjugated to horseradish peroxidase (Thermo Fisher Scientific; product no. 31430) at 1:1000 in blocking buffer. Samples were washed with blocking buffer three times and once with PBS. Plaques were detected using either TrueBlue peroxidase substrate (Seracare) according to the manufacturer's instructions (*Figure 8—figure supplement 2D–E*) or with ImmPACT DAB peroxidase substrate (Vector SK105) according to the manufacturer's instructions (*Figure 2C*). Images were captured using an EPSON scanner V600 at 1200 dpi. Plaque area (in pixels) was measured using Fiji by applying an intensity threshold to images of each plaque, creating binary masks with a value of 0 for background and 1 for plaque pixels (*Rueden et al., 2017*; *Schindelin et al., 2012*). Plaque area was determined by automated counting of each pixel within a plaque.

## CryoSIM

Cryopreserved TEM grids were placed onto a liquid nitrogen cryostage (Linkam) and were imaged by cryoSIM as previously described (*Vyas et al., 2021b*). The cryoSIM was developed in-house as previously described (*Phillips et al., 2020*). Hoechst stain fluorescence was excited using a 405 nm laser and was detected using an EM-452–45 filter (452±22.5 nm). eYFP-VP26 fluorescence was excited using a 488 nm laser and was detected using an EM-525–50 filter (525±25 nm). gM-mCherry fluorescence was excited using a 561 nm laser and was detected using an EM-605–70 filter (605±35 nm). MitoTracker Deep Red fluorescence was excited using a 647 nm laser and was detected using an EM-655-lp filter (≥655 nm). SIM data were reconstructed using SoftWoRx (Applied Precision Inc, Issaquah, WA) and the fluorescent channels were aligned using Chromagnon (*Matsuda et al., 2020*).

## CryoSXT

Cryopreserved TEM grids were loaded into a liquid nitrogen-cooled vacuum chamber of an UltraX-RM-S/L220c X-ray microscope (Carl Zeiss X-Ray Microscopy) at beamline B24 of the Diamond Light Source (*Harkiolaki et al., 2018*). Incident soft X-rays generated at the synchrotron (500 eV, $\lambda$ =2.48 nm) were used to illuminate the samples and transmitted X-rays were detected using a 1024B Pixis CCD camera (Princeton Instruments). Transmitted light was focused using a 25 nm zone plate objective with a nominal resolution limit of 25 nm. Samples were focused by Z translations of the zone plate and samples were centred along a rotational axis by Z translations of the sample grid. Individual X-ray projections (9.46×9.46 µm) were captured and tiled together in 7×7 montages known as X-ray mosaics to inspect sample quality and identify regions of interest for tomography. Tomographic data

were collected at fields of view measuring 9.46×9.46 μm. In each case, a collection of X-ray projections known as a tilt series were acquired by rotating the sample with maximum tilt angles of –60°/+60° and acquiring images at increments of 0.2° or 0.5°. A 0.5 or 1 s exposure time was used depending on the intensity of the transmitted X-rays. Tilt series were reconstructed using IMOD (version 4.9.2) (*Mastronarde and Held, 2017*) as previously described (*Nahas et al., 2022a*).

## Correlation of cryoSIM and cryoSXT

CryoSIM data was correlated onto CryoSXT data using easyCLEMv0 (*Paul-Gilloteaux et al., 2017*) as previously described (*Vyas et al., 2021a*) with a few differences. CryoSXT tomograms were used as the target for the correlation and a frame was added around the tomogram, increasing the XY dimensions to 1200×1200 voxels. This increased the canvas size of the transformed cryoSIM data and reduced the probability that edges of the cryoSIM data would overlap with the tomogram at the centre of the frame during XY translations and rotations of the cryoSIM data. To avoid the need for an affine transformation model, maximum Z projections of cryoSIM data were used to generate transformation files using the rigid transformation model. The X-ray mosaic and a minimum Z projection of the tomogram were used as targets for these transformations. Rigid-transformation files generated using the 2D maximum Z projections of the cryoSIM data were applied to the 3D Z stacks to prevent the anisotropic stretching of signal generated by affine transformations in lieu of rotation around the X or Y axis. Before conducting a 3D rigid transformation of cryoSIM Z stacks onto the target tomogram, the number of slices in the cryoSIM image was increased 5-fold (reducing the voxel depth from 125 to 25 nm). This made it easier to correlate the front and back edges of mitochondrial fluorescence and tomographic mitochondria together. The TransformJ plugin in Fiji was used to further fine-tune the correlation where needed (*Schindelin et al., 2012*).

## Quantitation of nuclear egress

3D CryoSIM images were collected from TEM grids containing U2OS cells infected at MOI = 2 with mutants as described in *Infection Assays*. A Hoechst 33342 stain was used to label the nucleus. Reconstructed images of eYFP-VP26, gM-mCherry, and Hoescht stain were registered together and maximum intensity projections in Z were generated using Fiji. Edges of the nucleus and plasma membranes were delineated using the Hoescht and gM-mCherry fluorescence, respectively, to determine nuclear and cytoplasmic regions of interest. For plasma membrane delineation, the gM-mCherry signal was oversaturated until a clear demarcation in signal intensity between the cell and the surrounding extracellular area was observed and used to delineate the cell boundary. Cells infected with the ΔgE virus were used as a negative control for attenuation in nuclear egress. As these cells did not contain a Hoescht stain, the borders of the nuclei in the X-ray mosaics were used to demarcate the nuclei in the fluorescence. A threshold was applied to the eYFP-VP26 maximum projection to filter out background and isolate pixels containing capsid fluorescence. The ratio of eYFP-VP26$^+$ pixels was measured between the nuclear and cytoplasmic regions of interest. A total of 95 infected cells were included in the analysis. Infected cells were included in the analysis if the JAC (determined by a concentration of gM-mCherry vesicles) was not located over or under the nucleus with respect to the XY plane. Exclusion criteria included cells with indistinguishable boundaries from each other, cells with a faint Hoescht nuclear stain, or cells that had a majority of their area located outside the field of view.

## Quantitation of cytoplasmic clustering

3D CryoSIM images were collected from TEM grids containing U2OS cells infected at MOI = 2 with mutants as described in *Infection Assays*. Reconstructed images of eYFP-VP26 and gM-mCherry were registered together using Chromagnon (*Matsuda et al., 2020*). Borders of the JAC were determined based on the relative intensity of the gM-mCherry fluorescence. Binary masks of eYFP-VP26 and gM-mCherry fluorescence were generated by applying thresholds that filter out background and noise. The total number of retained pixels in the borders of the JAC were counted for the two channels and an eYFP-VP26/gM-mCherry ratio was produced. In order to calculate colocalisation between these channels, a spatial coincidence-based method known as Manders correlation was used instead of intensity-based methods, such as Pearson's correlation, to minimize the influence of noise, background, and cryoSIM reconstruction artefacts. Thresholds on intensity were applied to the eYFP-VP26 and gM-mCherry fluorescent channels using Fiji, and the number of fluorescent pixels were

quantitated from masks (*Rueden et al., 2017*; *Schindelin et al., 2012*). Overlap between the channels was determined using the subtract function in Fiji. To calculate M1 values, masks of gM-mCherry fluorescence were subtracted from eYFP-VP26 masks, and the remaining pixels were quantitated and subtracted from the total pixels in the eYFP-VP26 mask to determine the number of eYFP-VP26 pixels that overlapped with gM-mCherry. M1 values were calculated in Excel (Microsoft) by dividing the number of overlapping pixels by the total number of eYFP-VP26 pixels in the region of interest. To calculate M2 values, the same process was performed but with the two channels switched. Exclusion criteria included cells where the JAC overlapped with the nucleus or cells where the JAC was partially excluded from the field of view.

## Quantitation of membrane intensity

The membrane intensity of vesicles surrounded by capsid arrays was measured from signed 16-bit tomograms generated by WBP without reducing noise by applying a SIRT-like filter. The minimum voxel intensity was measured from 3×3 voxel matrices at 30 points around the vesicle. Data were collected from three tandem projection planes spanning 30 nm in depth using the same 30 XY coordinates and the mean ± SD were graphed (*Figure 8E–H*). Of the 30 coordinates assessed, those nearest to proximal capsids were marked and this was used to draw a boundary between capsid-proximal and capsid-distal sides of the vesicle.

## Segmentation

Segmented volumes were generated manually using the Segmentation Editor in Fiji (*Schindelin et al., 2012*). The X-ray absorbance heatmap (*Figure 8G–H*) was generated by applying the Fire lookup table to the tomogram in Fiji and superimposing it onto the segmented volume of the vesicles (*Schindelin et al., 2012*). Segmented volumes were rendered in 3D using 3D Viewer in Fiji (*Schindelin et al., 2012*). Segmented vesicles (*Figure 7B–C*, *Figure 8A–D and I* and *Figure 8—figure supplement 2A*) appeared open-ended at the anterior and posterior faces with respect to the imaging plane because cryoSXT produced less contrast in those regions compared with side regions perpendicular to the imaging plane, making it impossible to fully segment the vesicles without extrapolation. This contrast discrepancy was not due to a limitation in the tomogram's depth as cryoSXT captured the entire volume of cells within each field of view but was due to a mechanical constraint in the X-ray microscope. During tilt series collection, the sample could only be rotated by a 120° range (–60° to +60°) rather than a desired 180° range (–90° to +90°) to avoid collisions with other components in the microscope. During acquisition, the vesicle's side regions were positioned parallel with the X-ray beam at 0°, maximising X-ray absorption and contrast. As the anterior and posterior faces perpendicular to the side regions were never positioned at ±90°, they were never parallel with the X-ray beam, reducing absorption and contrast in those regions. The open ends all share the same orientation, confirming that they are artefacts of the imaging setup rather than *bona fide* features.

## Graphs and statistics

Single-step replication curves (*Figure 2A*, *Figure 2—figure supplement 1*, and *Figure 8—figure supplement 2C*) and graphs of membrane intensity (*Figure 7E–F* and *Figure 7—figure supplement 1*) were generated in Excel (Microsoft). Relative plaque area graphs (*Figure 2B* and *Figure 8—figure supplement 2E*), the nuclear/cytoplasmic capsid ratio graph (*Figure 4B*), the eYFP-VP26/gM-mCherry ratio graph (*Figure 5D*), the Manders correlation graphs (*Figure 5E* and *Figure 5—figure supplement 1C*) and the graph showing the width of virus particle intermediates (*Figure 9*) were generated using the ggplot2 package *Gómez-Rubio, 2017* in R studio (*Racine, 2012*). The graph showing the width of vesicles surrounded by capsid arrays was generated using SuperPlots (*Lord et al., 2020*; *Figure 8—figure supplement 1A*). Single-step replication curves report the mean Log$_{10}$(PFU) values from two technical repeats and the error bars indicate range. For the graphs of membrane intensity for which the data was normally distributed, a two-tailed t-test was used to assess the significance of differences between voxel intensity between the capsid-proximal and capsid-distal sides of the vesicle (*Figure 8E–H*). Mann-Whitney *U* tests were used to assess significant differences for non-normally distributed data (*Figure 4B*, *Figure 5D–E*, *Figure 9*, *Figure 2B*, *Figure 5—figure supplement 1C*, and *Figure 8—figure supplement 2E*). The boundary was determined by the location of the capsids. The width of vesicles surrounded by capsid arrays was measured using *Contour* (*Nahas et al., 2022b*).

## Acknowledgements

We thank Diamond Light Source for access to beamline B24 (proposals MX18925, MX19958, BI21485, BI23508, BI25247, BI26657, and BI30442) and the experimental hall coordinators for helpful support. We thank members of beamline B24 at the Diamond Light Source (Thomas Fish, Archana Jadhav, Mohamed Koronfel, Ilias Kounatidis, Chidinma Okolo, Matt Spink, and Nina Vyas) for technical support with cryoSXT and cryoSIM. We thank the DNA Sequencing Facility at the Department of Biochemistry, University of Cambridge for their support. We thank Mike Hollinshead (University of Cambridge) for assistance with electron microscopy of HSV-1 infection. This work was supported by a PhD studentship co-funded by Diamond Light Source and the Department of Pathology, University of Cambridge, to KLN, by a Sir Henry Dale Fellowship, jointly funded by the Wellcome Trust and the Royal Society (098406/Z/12/B) to SCG, and by a Biotechnology and Biological Sciences Research Council (BBSRC) Research Grant (BB/M021424/1) to CMC. For the purpose of Open Access, the authors have applied a CC BY public copyright licence to any Author Accepted Manuscript (AAM) version arising from this submission.

## Additional information

### Funding

| Funder | Grant reference number | Author |
|---|---|---|
| Wellcome Trust | 10.35802/098406 | Stephen C Graham |
| Biotechnology and Biological Sciences Research Council | BB/M021424/1 | Colin M Crump |
| Diamond Light Source and the Department of Pathology, University of Cambridge | | Kamal L Nahas |

The funders had no role in study design, data collection and interpretation, or the decision to submit the work for publication. For the purpose of Open Access, the authors have applied a CC BY public copyright license to any Author Accepted Manuscript version arising from this submission.

### Author contributions

Kamal L Nahas, Conceptualization, Data curation, Software, Formal analysis, Validation, Investigation, Visualization, Methodology, Writing – original draft, Writing – review and editing; Viv Connor, Kaveesha J Wijesinghe, Henry G Barrow, Investigation; Ian M Dobbie, Resources, Software; Maria Harkiolaki, Conceptualization, Resources, Supervision, Funding acquisition, Investigation, Methodology, Project administration, Writing – review and editing; Stephen C Graham, Colin M Crump, Conceptualization, Supervision, Funding acquisition, Investigation, Writing – original draft, Project administration, Writing – review and editing

### Author ORCIDs

Kamal L Nahas ⓘ https://orcid.org/0000-0003-3501-8473
Henry G Barrow ⓘ https://orcid.org/0000-0001-6999-7812
Ian M Dobbie ⓘ https://orcid.org/0000-0002-5531-5865
Maria Harkiolaki ⓘ https://orcid.org/0000-0001-8091-9057
Stephen C Graham ⓘ https://orcid.org/0000-0003-4547-4034
Colin M Crump ⓘ https://orcid.org/0000-0001-9918-9998

Reviewer #1 (Public review): https://doi.org/10.7554/eLife.105209.3.sa1
Reviewer #2 (Public review): https://doi.org/10.7554/eLife.105209.3.sa2
Reviewer #3 (Public review): https://doi.org/10.7554/eLife.105209.3.sa3
Author response https://doi.org/10.7554/eLife.105209.3.sa4

## Additional files

### Supplementary files

Supplementary file 1. DNA oligonucleotide sequences. (A) Primers to generate mutants by two-step red recombination. Sequences homologous to the gene are in upper case, sequences homologous to the template plasmids pEP-KanS or pEP-RenillaLuc are in lower case, mutated start codons are underlined, and introduced stop codons are in bold. (B) PCR amplification primers for HSV-1 US3 and UL53. (C) Sanger sequencing primers for HSV-1 US3 and UL53.

MDAR checklist

### Data availability

Original imaging data for tomograms illustrated in the manuscript are available from the University of Cambridge Apollo Repository (https://doi.org/10.17863/CAM.121857). Recombinant virus strains generated for this project can be made available under a Material Transfer Agreement with the University of Cambridge – contact the corresponding author for requests.

The following dataset was generated:

| Author(s) | Year | Dataset title | Dataset URL | Database and Identifier |
| --- | --- | --- | --- | --- |
| Kamal N, Maria H, Stephen G, Crump CM | 2025 | Research data supporting: Applying 3D correlative structured illumination microscopy and X-ray tomography to characterise herpes simplex virus-1 morphogenesis | https://doi.org/10.17863/CAM.121857 | Apollo, 10.17863/CAM.121857 |

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
