## [Editor Report · eLife Assessment]

This **landmark** manuscript comprehensively examines the roles of nine structural proteins in herpes simplex virus 1 (HSV-1) assembly and nuclear egress. By integrating cryo-light microscopy and soft X-ray tomography, the study presents an innovative approach to investigating viral assembly within cells. The research is thoroughly executed, yielding **exceptional** data that explain previously unknown functions expected to bear widespread influence. This work is of broad interest to virologists, cellular biologists, and structural biologists, offering a robust, contextually rich methodology for studying large protein complex assembly within the cellular environment, serving as an excellent starting point for high-resolution techniques.

---

## [Referee Report · Reviewer #1 (Public review)]

Summary:

Nahas et al. investigated the roles of herpes simplex virus 1 (HSV-1) structural proteins using correlative cryo-light microscopy and soft X-ray tomography. The authors generated nine viral variants with deletions or mutations in genes encoding structural proteins. They employed a chemical fixation-free approach to study native-like events during viral assembly, enabling observation of a wider field of view compared to cryo-ET. The study effectively combined virology, cell biology, and structural biology to investigate the roles of viral proteins in virus assembly and budding.

Strengths:

(1) The study presented a novel approach to studying viral assembly in cellulo.

(2) The authors generated nine mutant viruses to investigate the roles of essential proteins in nuclear egress and cytoplasmic envelopment.

(3) The use of correlative imaging with cryoSIM and cryoSXT allowed for the study of viral assembly in a near-native state and in 3D.

(4) The study identified the roles of VP16, pUL16, pUL21, pUL34, and pUS3 in nuclear egress.

(5) The authors demonstrated that deletion of VP16, pUL11, gE, pUL51, or gK inhibits cytoplasmic envelopment.

(6) The manuscript is well-written, clearly describing findings, methods, and experimental design.

(7) The figures and data presentation are of good quality.

(8) The study effectively correlated light microscopy and X-ray tomography to follow virus assembly, providing a valuable approach for studying other viruses and cellular events.

(9) The research is a valuable starting point for investigating viral assembly using more sophisticated methods like cryo-ET with FIB-milling.

(10) The study proposes a detailed assembly mechanism and tracks the contributions of studied proteins to the assembly process.

(11) The study includes all necessary controls and tests for the influence of fluorescent proteins.

Weaknesses:

Overall, the manuscript does not have any major weaknesses, just a few minor comments, which were mostly solved in the revised version of the manuscript.

Comments on the latest version:

I reviewed the responses and the updated manuscript, and I am very pleased with how the authors have revised it. The manuscript was already strong, but with the addition of the summary table and the separated images, it is now excellent.

---

## [Referee Report · Reviewer #2 (Public review)]

Summary:

For centuries, humans have been developing methods to see ever smaller objects, such as cells and their contents. This has included studies of viruses and their interactions with host cells during processes extending from virion structure to the complex interactions between viruses and their host cells: virion entry, virus replication and virion assembly, and release of newly constructed virions. Recent developments have enabled simultaneous application of fluorescence-based detection and intracellular localization of molecules of interest in the context of sub-micron resolution imaging of cellular structures by electron microscopy.

The submission by Nahas et al., extends the state-of-the-art for visualization of important aspects of herpesvirus (HSV-1 in this instance) virion morphogenesis, a complex process that involves virus genome replication, and capsid assembly and filling in the nucleus, transport of the nascent nucleocapsid and some associated tegument proteins through the inner and outer nuclear membranes to the cytoplasm, orderly association of several thousand mostly viral proteins with the capsid to form the virion's tegument, envelopment of the tegumented capsid at a virus-tweaked secretory vesicle or at the plasma membrane, and release of mature virions at the plasma membrane.

In this groundbreaking study, cells infected with HSV-1 mutants that express fluorescently tagged versions of capsid (eYFP-VP26) and tegument (gM-mCherry) proteins were visualized with 3D correlative structured illumination microscopy and X-ray tomography. The maturation and egress pathways thus illuminated were studied further in infections with fluorescently tagged viruses lacking one of nine viral proteins.

Strengths:

This outstanding paper meets the journal's definitions of Landmark, Fundamental, Important, Valuable, and Useful. The work is also Exceptional, Compelling, Convincing, and Solid. The work is a tour de force of classical and state-of-the-art molecular and cellular virology. Beautiful images accompanied by appropriate statistical analyses and excellent figures. The numerous complex issues addressed are explained in a clear and coordinated manner; the sum of what was learned is greater than the sum of the parts. Impacts go well beyond cytomegalovirus and the rest of the herpesviruses, to other viruses and cell biology in general.

Comments on the latest version:

This is a very nice paper. The authors responded affirmatively to the suggestions and questions of the reviewers.

---

## [Referee Report · Reviewer #3 (Public review)]

Summary:

Kamal L. Nahas et al. demonstrated that pUL16, pUL21, pUL34, VP16, and pUS3 are involved in the egress of the capsids from the nucleous, since mutant viruses ΔpUL16, ΔpUL21, ΔUL34, ΔVP16, and ΔUS3 HSV-1 show nuclear egress attenuation determined by measuring the nuclear:cytoplasmic ratio of the capsids, the dfParental, or the mutants. Then, they showed that gM-mCherry+ endomembrane association and capsid clustering were different in pUL11, pUL51, gE, gK, and VP16 mutants. Furthermore, the 3D view of cytoplasmic budding events suggests an envelopment mechanism where capsid budding into spherical/ellipsoidal vesicles drives the envelopment.

Strengths:

The authors employed both structured illumination microscopy and cellular ultrastructure analysis to examine the same infected cells, using cryo-soft-X-ray tomography to capture images. This combination, set here for the first time, enabled the authors to obtain holistic data regarding a biological process, as a viral assembly. Using this approach, the researchers studied various stages of HSV-1 assembly. For this, they constructed a dual-fluorescently labelled recombinant virus, consisting of eYFP-tagged capsids and mCherry-tagged envelopes, allowing for the independent identification of both unenveloped and enveloped particles. They then constructed nine mutants, each targeting a single viral protein known to be involved in nuclear egress and envelopment in the cytoplasm, using this dual-fluorescent as the parental one. The experimental setting, both the microscopic and the virological, is robust and well-controlled. The manuscript is well-written, and the data generated is robust and consistent with previous observations made in the field.

I congratulate the authors. The work is robust, and I personally highlight the way they managed to include others' results merged among their own, providing a complete view of the story.

Comments on the latest version:

I reviewed the responses and the updated manuscript, and I agree with the reviewer's #1 words: "The manuscript was already strong, but with the addition of the summary table and the separated images, it is now excellent."

---

## [Author Response]

The following is the authors’ response to the original reviews

**Public Reviews:**

**Reviewer #1 (Public review):**
Summary:Nahas et al. investigated the roles of herpes simplex virus 1 (HSV-1) structural proteins using correlative cryo-light microscopy and soft X-ray tomography. The authors generated nine viral variants with deletions or mutations in genes encoding structural proteins. They employed a chemical fixation-free approach to study native-like events during viral assembly, enabling observation of a wider field of view compared to cryo-ET. The study effectively combined virology, cell biology, and structural biology to investigate the roles of viral proteins in virus assembly and budding.Strengths:(1) The study presented a novel approach to studying viral assembly in cellulo.(2) The authors generated nine mutant viruses to investigate the roles of essential proteins in nuclear egress and cytoplasmic envelopment.(3) The use of correlative imaging with cryoSIM and cryoSXT allowed for the study of viral assembly in a near-native state and in 3D.(4) The study identified the roles of VP16, pUL16, pUL21, pUL34, and pUS3 in nuclear egress.(5) The authors demonstrated that deletion of VP16, pUL11, gE, pUL51, or gK inhibits cytoplasmic envelopment.(6) The manuscript is well-written, clearly describing findings, methods, and experimental design.(7) The figures and data presentation are of good quality.(8) The study effectively correlated light microscopy and X-ray tomography to follow virus assembly, providing a valuable approach for studying other viruses and cellular events.(9) The research is a valuable starting point for investigating viral assembly using more sophisticated methods like cryo-ET with FIB-milling.(10) The study proposes a detailed assembly mechanism and tracks the contributions of studied proteins to the assembly process.(11) The study includes all necessary controls and tests for the influence of fluorescent proteins.Weaknesses:Overall, the manuscript does not have any major weaknesses, just a few minor comments:(1) The gel quality in Figure 1 is inconsistent for different samples, with some bands not well resolved (e.g., for pUL11, GAPDH, or pUL20).

We thank the reviewer for their suggestion. We tried to resolve the bands several times, but unfortunately this was the best outcome we could achieve.

(2) The manuscript would benefit from a summary figure or table to concisely present the findings for each protein. It is a large body of manuscript, and a summary figure showing the discovered function would be great.

We thank the reviewer for their suggestion. We have created a summary table (Table 2).

(3) Figure 2 lacks clarity on the type of error bars used (range, standard error, or standard deviation). It says, however, range, and just checking if this is what the authors meant.

We thank the reviewer for double-checking, but it is meant to be range, as reported in the legend. We used range because there are only two data points for each time point, which are insufficient to calculate standard deviation or standard error.

(4) The manuscript could be improved by including details on how the plasma membrane boundary was estimated from the saturated gM-mCherry signal. An additional supplementary figure with the data showing the saturation used for the boundary definition would be helpful.

We appreciate the suggestion and have included an example of how saturated gM-mCherry signal was used to delineate the cytoplasm in Supp. Fig. 4A.

(5) Additional information or supplementary figures on the mask used to filter the YFP signal for Figure 4 would be helpful.

Thanks, we have adapted the text in the results section to clarify: “eYFP-VP26 signal was manually inspected to determine threshold values that filtered out background and included pixels containing individual or clustered puncta that represent capsids.”

(6) The figure legends could include information about which samples are used for comparison for significance calculations. As the colour of the brackets is different from the compared values (dUL34), it would be great to have this information in the figure legend.

Thanks, we have adapted Fig. 4B to make the colour of the brackets match the colour used for the ΔUL34 mutant, and we have included labels next to the brackets for clarity. We have applied similar adjustments to Fig. 5D & E and Supp. Fig. 4C.

(7) In Figure 5B, the association between YFP and mCherry signals is difficult to assess due to the abundance of mCherry signal; single-channel and combined images might improve visualization.

Thanks, we have provided split and combined channel views in Supp. Fig. 4B to improve visualization.

(8) In Figure 6D, staining for tubulin could help identify the cytoskeleton structures involved in the observed virus arrays.

We thank the reviewer for their suggestion, which we think would be interesting future work to build on the current study. Given the competitive nature of access to the cryoSIM and cryoSXT, CLXT, including staining for tubulin was outside the scope of additional experiments we were able to conduct at this time.

(9) It is unclear in Figure 6D if the microtubule-associated capsids are with the gM envelope or not, as the signal from mCherry is quite weak. It could be made clearer with the split signals to assess the presence of both viral components.

We have provided split channels to the figure to aid with visualization.

(10) The representation of voxel intensity in Figure 8 is somewhat confusing. Reversion of the voxel intensity representation to align brighter values with higher absorption, which would simplify interpretation.

We thank the reviewer for this suggestion. In contrast to fluorescence microscopy where high intensities reflect signal, low intensities represent signal (absorbance of X-rays) in cryoSXT. We respectfully decided not to reverse the values, as we believe that could cause more confusion. We have instead added a black-to-white gradient bar to illustrate that low voxel intensities correspond to dark signal in Fig 8.

(11) The visualization in panel I of Figure 8 might benefit from a more divergent colormap to better show the variation in X-ray absorbance.

We thank the reviewer for their suggestion. We experimented with a few different colour schemes but concluded that the current one produced the clearest results and was most accessible for color-blind viewers.

(12) Figure 9 would be enhanced by images showing the different virus sizes measured for the comparative study, which would help assess the size differences between different assembly stages.

We thank the reviewer for their suggestion and have included images to accompany the graph.

Overall, this is an excellent manuscript and an enjoyable read. It would be interesting to see this approach applied to the study of other viruses, providing valuable insights before progressing to high-resolution methods.
**Reviewer #2 (Public review):**
Summary:For centuries, humans have been developing methods to see ever smaller objects, such as cells and their contents. This has included studies of viruses and their interactions with host cells during processes extending from virion structure to the complex interactions between viruses and their host cells: virion entry, virus replication and virion assembly, and release of newly constructed virions. Recent developments have enabled simultaneous application of fluorescence-based detection and intracellular localization of molecules of interest in the context of sub-micron resolution imaging of cellular structures by electron microscopy.The submission by Nahas et al., extends the state-of-the-art for visualization of important aspects of herpesvirus (HSV-1 in this instance) virion morphogenesis, a complex process that involves virus genome replication, and capsid assembly and filling in the nucleus, transport of the nascent nucleocapsid and some associated tegument proteins through the inner and outer nuclear membranes to the cytoplasm, orderly association of several thousand mostly viral proteins with the capsid to form the virion's tegument, envelopment of the tegumented capsid at a virus-tweaked secretory vesicle or at the plasma membrane, and release of mature virions at the plasma membrane.In this groundbreaking study, cells infected with HSV-1 mutants that express fluorescently tagged versions of capsid (eYFP-VP26) and tegument (gM-mCherry) proteins were visualized with 3D correlative structured illumination microscopy and X-ray tomography. The maturation and egress pathways thus illuminated were studied further in infections with fluorescently tagged viruses lacking one of nine viral proteins.Strengths:This outstanding paper meets the journal's definitions of Landmark, Fundamental, Important, Valuable, and Useful. The work is also Exceptional, Compelling, Convincing, and Solid. The work is a tour de force of classical and state-of-the-art molecular and cellular virology. Beautiful images accompanied by appropriate statistical analyses and excellent figures. The numerous complex issues addressed are explained in a clear and coordinated manner; the sum of what was learned is greater than the sum of the parts. Impacts go well beyond cytomegalovirus and the rest of the herpesviruses, to other viruses and cell biology in general.
**Reviewer #3 (Public review):**
Summary:Kamal L. Nahas et al. demonstrated that pUL16, pUL21, pUL34, VP16, and pUS3 are involved in the egress of the capsids from the nucleous, since mutant viruses ΔpUL16, ΔpUL21, ΔUL34, ΔVP16, and ΔUS3 HSV-1 show nuclear egress attenuation determined by measuring the nuclear:cytoplasmic ratio of the capsids, the dfParental, or the mutants. Then, they showed that gM-mCherry+ endomembrane association and capsid clustering were different in pUL11, pUL51, gE, gK, and VP16 mutants. Furthermore, the 3D view of cytoplasmic budding events suggests an envelopment mechanism where capsid budding into spherical/ellipsoidal vesicles drives the envelopment.Strengths:The authors employed both structured illumination microscopy and cellular ultrastructure analysis to examine the same infected cells, using cryo-soft-X-ray tomography to capture images. This combination, set here for the first time, enabled the authors to obtain holistic data regarding a biological process, as a viral assembly. Using this approach, the researchers studied various stages of HSV-1 assembly. For this, they constructed a dual-fluorescently labelled recombinant virus, consisting of eYFP-tagged capsids and mCherry-tagged envelopes, allowing for the independent identification of both unenveloped and enveloped particles. They then constructed nine mutants, each targeting a single viral protein known to be involved in nuclear egress and envelopment in the cytoplasm, using this dual-fluorescent as the parental one. The experimental setting, both the microscopic and the virological, is robust and well-controlled. The manuscript is well-written, and the data generated is robust and consistent with previous observations made in the field.Weaknesses:It would be helpful to find out what role the targeted proteins play in nuclear egress or envelopment acquisition in a different orthoherpesvirus, like HSV-2. This would confirm the suitability of the technical approach set and would also act as a way to validate their mechanism at least in one additional herpesvirus beyond HSV-1. So, using the current manuscript as a starting point and for future studies, it would be advisable to focus on the protein functions of other viruses and compare them.

We appreciate the suggestion and agree that this would be a great starting point for future studies. At present, we do not have a panel of mutant viruses in HSV-2 or another orthoherpesvirus, and it would be significant work to generate them, so we consider this outside the scope of the current study.

**Recommendations for the authors:**

**Reviewer #2 (Recommendations for the authors):**
(1) There are enough uncommon abbreviations in the text to justify the inclusion of an abbreviation list.

We thank the reviewer for the suggestion, but we define all uncommon abbreviations at first mention and an abbreviations list is not part of eLife’s house style.

(2) The complex paragraph on p. 7 would be much easier to digest if broken into smaller chunks. Consider similar treatment for other lengthy landmark-free blocks of text, e.g., the one that begins on p. 14. Subheadings would help.

We thank the reviewer for this suggestion. We have divided large paragraphs into more easily digestible chunks throughout the manuscript, for example in the discussion where the previous monolithic 3rd paragraph has been divided into five shorter, focussed paragraphs.

(3) Table 1 needs units.

We thank the reviewer for noticing our omission and apologise for the oversight - the table has been updated accordingly.

**Reviewer #3 (Recommendations for the authors):**
(1) Toward the end of the manuscript, I missed some lines attempting to speculate on the origin/nature of the spherical/ellipsoidal vesicles providing the envelopment. Would it be possible to incorporate this in the Discussion section?

Thank you for noticing that omission. We have now included a few lines speculating that they may represent recycling endosomes, *trans*-Golgi network vesicles, or a hybrid compartment.

(2) I congratulate the authors. The work is robust, and I personally highlight the way they managed to include others' results merged with their own, providing a complete view of the story.

We thank the reviewer for their kind words.

**Note to editors**

In addition to these responses to the reviewer’s comments, we have also now included in the methods section details of the Tracking of Indels by Decomposition (TIDE) analysis we performed (data in Supplementary Figure 3) that was omitted by mistake from the original submission.